# Self-assembly of an anion receptor with metal-dependent kinase inhibition and potent in vitro anti-cancer properties

Simon J. Allison [1✉], Jaroslaw Bryk[1], Christopher J. Clemett [1], Robert A. Faulkner [1], Michael Ginger[1], Hollie B. S. Griffiths[1], Jane Harmer [1], P. Jane Owen-Lynch [1], Emma Pinder [1], Heiko Wurdak[2], Roger M. Phillips[1✉] & Craig R. Rice [1✉]

One topical area of supramolecular chemistry is the binding of anionic species but despite the importance of anions in diverse cellular processes and for cancer development, anion receptors or 'binders' have received little attention as potential anti-cancer therapeutics. Here we report self-assembling trimetallic cryptands (e.g. $[\mathbf{L}_2(\text{Metal})_3]^{6+}$ where Metal = $Cu^{2+}$, $Zn^{2+}$ or $Mn^{2+}$) which can encapsulate a range of anions and which show metal-dependent differences in chemical and biological reactivities. In cell studies, both $[\mathbf{L}_2Cu_3]^{6+}$ and $[\mathbf{L}_2Zn_3]^{6+}$ complexes are highly toxic to a range of human cancer cell lines and they show significant metal-dependent selective activity towards cancer cells compared to healthy, non-cancerous cells (by up to 2000-fold). The addition of different anions to the complexes (e.g. $PO_4^{3-}$, $SO_4^{2-}$ or $PhOPO_3^{2-}$) further alters activity and selectivity allowing the activity to be modulated via a self-assembly process. The activity is attributed to the ability to either bind or hydrolyse phosphate esters and mechanistic studies show differential and selective inhibition of multiple kinases by both $[\mathbf{L}_2Cu_3]^{6+}$ and $[\mathbf{L}_2Zn_3]^{6+}$ complexes but via different mechanisms.

[1] School of Applied Sciences, University of Huddersfield Queensgate, Huddersfield, UK. [2] School of Medicine, University of Leeds, Leeds, UK. ✉email: s.allison@hud.ac.uk; r.m.phillips@hud.ac.uk; c.r.rice@hud.ac.uk

Contemporary medicines range from the relatively simple such as lithium salts for the treatment of bipolar disorder, to large and complex organic structures for cancer therapy, which can contain hundreds of atoms and dozens of chiral centres[1]. However, despite their large diversity they all share a commonality; *viz* all are discrete molecular species that have to be chemically prepared often via an iterative synthetic procedure. A different approach to this is the use of self-assembly, which is a process where a disordered system of pre-existing compounds forms an organised structure as a consequence of specific pre-programmed interactions among the components themselves[2–9]. Self-assembly offers easy and rapid access to a library of molecularly complex architectures and novel compounds all of which may have differing biological activity without the need for a chemist to conventionally synthesise the differing molecules.

To date artificial self-assembling systems have received relatively little attention for potential therapeutic applications despite self-assembly becoming a mature area of scientific study[10–14]. However, there are some notable advances in this area, for example Hannon and co-workers have shown that a dinuclear triple helicate (e.g. $[L_3Fe_2]^{4+}$) interacts strongly with duplex DNA, and displays both anti-cancer and anti-bacterial properties[15–17]. Scott and co-workers have demonstrated that Fe(II)-containing "head-to-head-to-tail" helicates show toxic and selective in vitro cytotoxic activity against a range of cancer cell lines with $IC_{50}$ values lower than *cis*-platin[18]. Ruthenium containing transition metal helicates and mesocates have also been shown to have interesting biological properties with some showing higher cytotoxicity towards cancer cells that lack p53[19]. Metallacycles and metallacages have also been shown to possess useful properties with HHan's $[Pd_2L_4]^{4+}$ cages showing cytotoxicity towards an array of different human cancer cell lines[20]. Unsurprisingly, many of these active assemblies contain $Pt^{2+}$ as its cytotoxic effects are well known and these have been shown to have activity towards a number of cancer cell lines, with some assemblies used to encapsulate and target *cis*-platin delivery[21].

The binding of anionic species, for example by synthetic scaffolds containing hydrogen-bond donor units, is a topical area of supramolecular chemistry[22–27]. Due to the importance of different anions in diverse biological processes it would seem highly likely that anion binders would have interesting biological effects and potentially could have valuable therapeutic properties where anion-dependent processes are perturbed[28]. Indeed, recent results have shown that synthetic chloride transporters can compensate for insufficient chloride fluxes in cystic fibrosis however, any potential anti-cancer properties of anion receptors have yet to be explored[29,30]. We have previously demonstrated that the tripodal ligand L, which contains three bidentate *N*-donor units separated by a tris(2-aminoethyl) amine spacer, forms a cryptand with $Cu^{2+}$ (e.g. $[L_2Cu_3]^{6+}$). These assembles can bind a large variety of anions of different geometries and the complex can efficiently remove phosphate anions from aqueous media, reducing levels from 1000 to <0.1 ppm, even in the presence of competitive species[31].

In this work we demonstrate that the tripodal ligand L also self-assembles with both $Zn^{2+}$ and $Mn^{2+}$ ions to form a trinuclear species (e.g. $[L_2Zn_3]^{6+}$) and in these anions are encapsulated both in the solid-state and aqueous systems (e.g. $[L_2M_3(PO_4)]^{3+}$ and $[L_2M_3(SO_4)]^{4+}$ where M = $Zn^{2+}$ or $Mn^{2+}$). Interestingly, these metal assemblies (M = $Zn^{2+}$, $Mn^{2+}$ or $Cu^{2+}$) show different chemical reactivities towards a phenyl phosphate dianion ($PhOPO_3^{2-}$) with the copper-containing species incorporating the anion in the assembly (e.g. $[L_2Cu_3(O_3POPh)]^{4+}$) whereas the $Mn^{2+}$ and $Zn^{2+}$ complexes hydrolyse the anion to phosphate at different rates (e.g. $[L_2M_3(PO_4)]^{3+}$ where M = $Zn^{2+}$ or $Mn^{2+}$) indicating intrinsic phosphatase activity of these complexes (Fig. 1). In cell studies, both $[L_2Cu_3]^{6+}$ and $[L_2Zn_3]^{6+}$ complexes

are highly toxic to a range of human cancer cell lines as well as a glioblastoma cancer stem cell model[32,33]. As well as potency, both complexes show high selective activity towards cancer cells compared to healthy, non-cancerous cells (by up to 2000-fold; ARPE-19[34], MCF10A[35], and NP1[32,33] non-cancer cell models). Encapsulation of either a phosphate or sulfate anion further modulates potency and selectivity. In contrast to $Zn^{2+}$ and $Cu^{2+}$ complexes, $[L_2Mn_3]^{6+}$ showed only modest selective activity which was comparable to that of clinically used platinate anticancer agents. Mechanism of action studies show that $[L_2Zn_3]^{6+}$ has selective phosphatase activity towards phospho-serine, phospho-tyrosine and phospho-threonine amino acids resulting in the selective inhibition of multiple kinases, cancer cell ATP depletion, autophagy and cancer cell toxicity.

## Results

**Synthesis and characterisation.** Reaction of 1.5 equivalents of L with either $Zn(ClO_4)_2$ or $Mn(ClO_4)_2$ results in a mononuclear complex (e.g. $[LM]^{2+}$) as demonstrated by X-ray crystallography and ESI-MS. In the solid-state the zinc complex contains a 6-coordinate $Zn^{2+}$ cation coordinated by six nitrogen atoms from three pyridyl-thiazole bidentate units from the same ligand. In the $Mn^{2+}$ analogue the metal ion is again 6-coordinate, but this arises from coordination by four *N*-donor atoms from two bidentate pyridyl-thiazole units and two water *O*-donor atoms (Fig. 2a, b). Both complexes differ from the $Cu^{2+}$ derivative which can form the trimetallic capsule (e.g. $[L_2Cu_3]^{6+}$) even in the presence of weakly interacting anions. The difference is attributed to the ability of $Cu^{2+}$ to form 4-coordinate complexes (at least with pyridyl-thiazole donors) whereas both $Zn^{2+}$ and $Mn^{2+}$ prefer higher-coordinate geometry and without a strongly coordinating anion present a simple mononuclear species is formed. The coordination of water in the $Mn^{2+}$ complex (e.g. $[LMn(H_2O)_2]^{2+}$) is a consequence of the oxophillic nature of this hard cation and this behaviour is mirrored in all the structures with this metal. The formation of these species is also observed in the gas phase with ESI-MS studies showing that only ions corresponding to the mononuclear species are present.

Reaction of L with either $Mn^{2+}$ or $Zn^{2+}$ with $(Bu_4N)HSO_4$ (in the correct stoichiometric proportions) results in the formation of the capsule in which sulfate anions are encapsulated (e.g. $[L_2M_3(SO_4)]^{4+}$) (Fig. 2c–f). In the solid state the $Zn^{2+}$ is isostructural to the $Cu^{2+}$ derivative with a trinuclear $[L_2Zn_3]^{6+}$ assembly and within it is an encapsulated sulfate anion (e.g. $[L_2Zn_3(SO_4)]^{4+}$). Each of the three $Zn^{2+}$ atoms are 5-coordinate arising from four *N*-donor atoms from two bidentate pyridyl-thiazole units and one oxygen donor from the sulfate anion. The sulfate is held within the capsule by three coordination bonds to $Zn^{2+}$ supplemented by three -NH···O hydrogen bonding interactions. The remaining uncoordinated oxygen atom forms hydrogen bonds to three -NH donor units within the "upper rim" of the cavity. The $Mn^{2+}$ forms a very similar type of assembly but each of the $Mn^{2+}$ metal ions are 6-coordinate, which for one of the metal ions arises from four *N*-donor atoms from two bidentate pyridyl-thiazole units and two oxygen donors from the sulfate anion. The remaining two ions are also 6-coordinate and are coordinated by two bidentate *N*-donor ligands but are coordinated by one oxygen atom from the sulfate anion and one water molecule (e.g. $[L_2Mn_3(H_2O)_2(SO_4)]^{4+}$). This demonstrates that whilst with weakly interacting anions (e.g. halides, $ClO_4^-$, and $BF_4^-$) both $Zn^{2+}$ and $Mn^{2+}$ form mononuclear complexes, with tetrahedral oxoanions these template the formation of the trimetallic capsule.

Ions in the ESI-MS at *m/z* 1844 and 1812 corresponding to $\{[L_2Zn_3(SO_4)(ClO_4)_3]\}^+$ and $\{[L_2Mn_3(SO_4)(ClO_4)_3]\}^+$ coupled

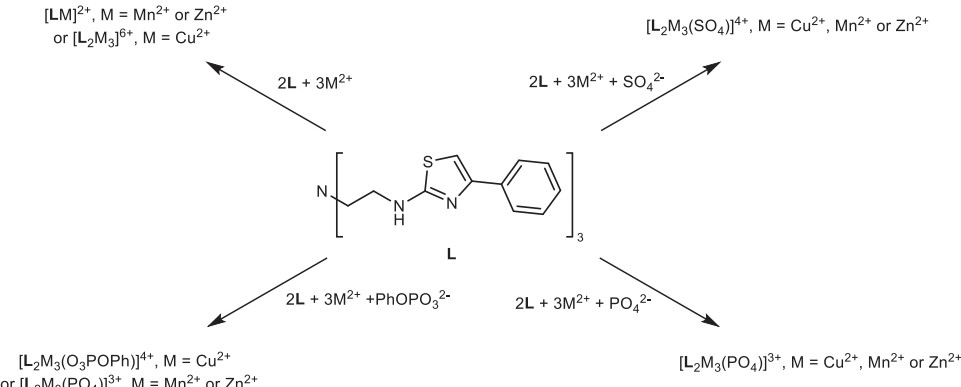

**Fig. 1 Differing complexes formed with ligand L, transition metal ions and different anions.** Trimetallic assemblies are formed between $Cu^{2+}$ and L whereas mononuclear species result upon reaction of either $Zn^{2+}$ or $Mn^{2+}$. Trinuclear assemblies can be formed with the latter two cations by templation by $SO_4^{2-}$ or $PO_4^{3-}$. $PhOPO_3^{2-}$ is encapsulated by the copper complex but is hydrolysed to $PO_4^{3-}$ when either the $Zn^{2+}$ or $Mn^{2+}$ are used. Reaction of the ligand **L** with $Cu^{2+}$, $Zn^{2+}$ and $Mn^{2+}$ and various anions.

with doubly charged ions indicate that these species are also observed in the gas phase.

Addition of disodium phenylphosphate to a solution of $[L_2Cu_3]^{6+}$ in MeCN/$H_2O$ results in a colour change from light blue to green. Crystals were then deposited after several days and analysis by X-ray crystallography shows that the trimetallic capsule is still formed but held inside the host is a $PhOPO_3^{2-}$ anion. In a very similar fashion to the other oxoanions, $PhOPO_3^{2-}$ is coordinated to the three $Cu^{2+}$ metal ions supplemented by a series of -NH⋯anion interactions. However, due to the phenyl substituent the ligands adopt a slightly different conformation allowing the phenyl unit to occupy a cleft formed by two pyridyl-thiazole units (Fig. 3a, b).

Reaction of two equivalents of **L**, three equivalents of either M $(ClO_4)_2$ (where M = $Zn^{2+}$ or $Mn^{2+}$) and $PhOPO_3Na_2$ results in a very different species. In the solid-state both structures contain a central $PO_4^{3-}$ anion held within the molecule by a series of interactions between the metal ions and amine hydrogen atoms (Fig. 3c–f). The $[L_2Mn_3(PO_4)]^{3+}$ complex is similar to the sulfate analogue and the three 6-coordinate $Mn^{2+}$ metal ions are coordinated by two bidentate N-donor ligand domains but one metal ion is coordinated by two oxygen atoms from the anion and the remaining two metal ions are coordinated by one anion oxygen atom and a water molecule. The $[L_2Zn_3(PO_4)]^{3+}$ is slightly different from the sulfate analogue and one metal ion is 6-coordinate arising from coordination of two bidentate N-donor ligand units and two oxygen atoms of the anion. The remaining two metal ions are only 5-coordinate as only one oxygen atom from the anion interacts with the metal.

This metal-dependent reactivity is also observed in the ESI-MS. Reaction of $[L_2Cu_3](ClO_4)_6$ with $PhOPO_3Na_2$ in water and MeCN gives ions at *m/z* 1914, 1024, 788 and 463 corresponding to $\{[L_2Cu_3(PhOPO_3)](ClO_4)_3\}^+$, $\{[LCu_2(PhOPO_3)](ClO_4)\}^+$, $\{[LCu](ClO_4)\}^+$ and $\{[LCu_2(PhOPO_3)]\}^{2+}$. Heating this sample at 80 °C shows no change in the ESI-MS spectrum indicating that the phenylphosphate dianion remains intact. A similar reaction of $PhOPO_3Na_2$ with $Zn(ClO_4)_2$ and **L** gave an ESI-MS with ions at *m/z* 1920 and 910 corresponding to $\{[L_2Zn_3(PhOPO_3)](ClO_4)_3\}^+$ and $\{[L_2Zn_3(PhOPO_3)](ClO_4)_2\}^{2+}$ respectively. Lower molecular weight ions at m/z 1029, 791 and 464 corresponding to $\{[LZn_2(PhOPO_3)](ClO_4)\}^+$, $\{[LZn](ClO_4)\}^+$ and $\{[LZn_2(PhOPO_3)]\}^{2+}$ were also observed. However, heating this sample at 80 °C results in a dramatic change in the ESI-MS with the spectrum now much simplified with ions at *m/z* 1743 and 822 corresponding to $\{[L_2Zn_3(PO_4)](ClO_4)_2\}^+$ and $\{[L_2Zn_3(PO_4)](ClO_4)\}^{2+}$. This demonstrates that initially the $Zn^{2+}$ containing complex reacts with phenyl phosphate dianion and in a similar fashion to the $Cu^{2+}$ analogue

and forms the trinuclear complex incorporating this anion (e.g. $[L_2Zn_3(PhOPO_3)]^{4+}$). However, after either a few days at room temperature or heating at 80 °C for 1 h the anion is hydrolysed and phosphate is encapsulated within the cryptand (see Supplementary Information). This hydrolysis is also confirmed by [1]H NMR as reaction of two equivalents of **L**, three equivalents of $Zn(ClO_4)_2$ and $PhOPO_3Na_2$ initially gives a broad complex spectrum but after 1 h at 80 °C gave a spectrum that contains signals corresponding to $[L_2Zn_3(PO_4)]^{3+}$ accompanied with signals corresponding to the phenol hydrolysis product. Reaction with $Mn(ClO_4)_2$ is similar to the $Zn^{2+}$ analogue with ions in the ESI-MS corresponding to binding of phenyl phosphate observed initially (e.g. $\{[LMn_2(PhOPO_3)](ClO_4)\}^+$) but ions corresponding to hydrolysis (e.g. $\{[L_2Mn_3(PO_4)](ClO_4)_2\}^+$ and $\{[L_2Mn_3(PO_4)](ClO_4)\}^{2+}$) are observed after heating for 1 h[36].

The $Zn^{2+}$ complex shows substrate specific differences in the rates of hydrolysis and its phosphatase activity. Analysis of the hydrolysis of phenyl phosphate dianion by the $Zn^{2+}$ complex (in a 25%:75% mixture of DMSO and buffered $H_2O$ solution (HEPES 60 mmol, *p*H 7.5)) by [31]P NMR shows a signal at −1.5 ppm corresponding to unhydrolysed $PhOPO_3^{2-}$ at *t* = 0 (Fig. 4, spectra b). After 19 h at 37 °C the major signal present is now observed at 8.6 ppm which is at an identical chemical shift to $[L_2Zn_3(PO_4)]^{3+}$ and after 44 h virtually no signal corresponding to $PhOPO_3^{2-}$ is observed (Fig. 4, spectra c, d). In a similar experiment using 4-nitrophenyl phosphate no [31]P signals could be detected that corresponded to the starting material following mixing with the $Zn^{2+}$ complex indicating almost immediate hydrolysis. Only $[L_2Zn_3(PO_4)]^{3+}$ was observed, coupled with a rapid yellowing of the solution due to the formation of 4-nitrophenolate.

Given the importance of protein phosphorylation in inter- and intra- cellular signalling and to cell function, and its common dysregulation in cancers[37,38], it was next analysed whether the $Zn^{2+}$ complex could dephosphorylate the phosphorylated amino acids serine, threonine and tyrosine. Indeed, the $Zn^{2+}$ complex resulted in dephosphorylation of serine-$OPO_3^{2-}$ and tyrosine-$OPO_3^{2-}$ at similar hydrolysis rates to $PhOPO_3^{2-}$ with substantial hydrolysis occurring over 24 h and completion after 48 h (Fig. 4, spectra e–h; m–p). Dephosphorylation of the amino acid threonine-$OPO_3^{2-}$ by the $Zn^{2+}$ complex was much slower however and after 48 h, threonine-$OPO_3^{2-}$ was still the major species (Fig. 4, spectra i–l). Differences in reactivity towards different substrates can be attributed to both steric and electronic effects. The difference in reactivity towards phenyl phosphate compared to the 4-nitro derivative is likely a consequence of the

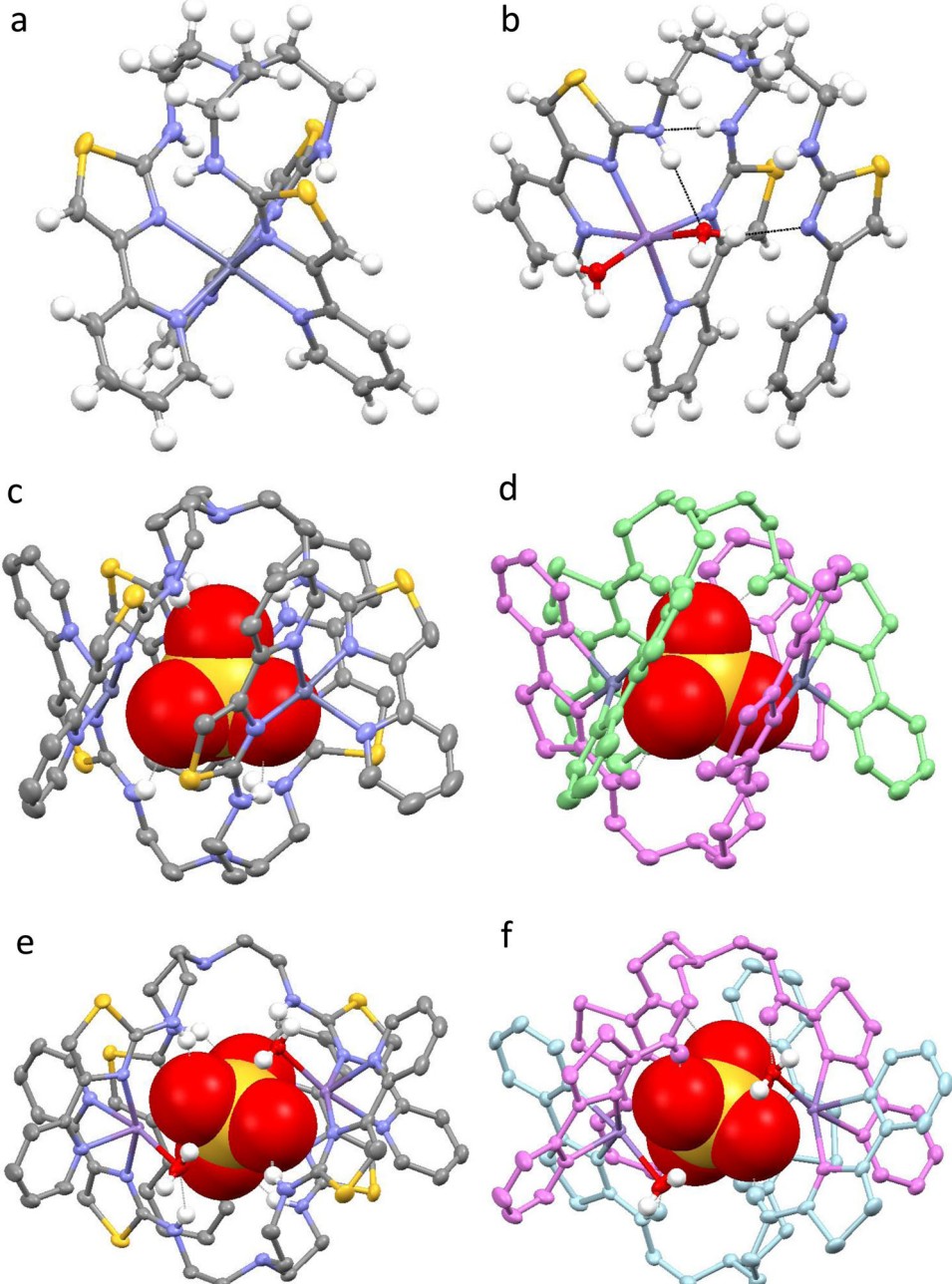

**Fig. 2 Single-crystal X-ray structures of metal-containing complexes of L. a** [LZn]$^{2+}$. **b** [LMn(H$_2$O)$_2$]$^{2+}$. **c** [L$_2$Zn$_3$(SO$_4$)]$^{4+}$. **d** [L$_2$Zn$_3$(SO$_4$)]$^{4+}$. **e** [L$_2$Mn$_3$(H$_2$O)$_2$(SO$_4$)]$^{4+}$. **f** [L$_2$Mn$_3$(H$_2$O)$_2$(SO$_4$)]$^{4+}$. Colour code: dark blue, Zn(II); purple, Mn; red, O; blue, N; yellow, S; grey, C (apart from **2d** and **2f** where the ligand strands have been coloured for clarity).

electron-withdrawing nitro group, which will enhance the hydrolysis. Serine-PO$_3$$^{2-}$, tyrosine-OPO$_3$$^{2-}$ and threonine-OPO$_3$$^{2-}$ all have similar electronic properties, but threonine has a methyl substituent close to the phosphorylated residue and it would seem likely this would result in unfavourable steric interactions upon binding of [L$_2$Zn$_3$]$^{6+}$ as the −CHCH$_3$ unit would be housed deep in the cleft of the self-assembled species (see Fig. 3a, b). It seems probable that this interaction would reduce the ability of the cryptand to bind the anion and hence reduce the hydrolysis rate. Both serine and tyrosine are less sterically demanding (tyrosine-OPO$_3$$^{2-}$ is very similar to PhOPO$_3$$^{2-}$ and serine-PO$_3$$^{2-}$ has a less sterically demanding −CH$_2$− unit in this position) and consequently are hydrolysed

more rapidly. In all cases, in a similar fashion to that observed with phenyl phosphate (Supplementary Fig. 4.2), we presume the phosphorylated amino acids are hydrolysed to the corresponding alcohol.

The rate of substrate hydrolysis is also dependent upon the metal used in the self-assembly process. It is clear from the solid-state and ESI-MS data that [L$_2$Cu$_3$]$^{6+}$ does not hydrolyse phenyl phosphate but incorporates this anion within the assembly e.g. [L$_2$Cu$_3$(PhOPO$_3$)]$^{4+}$. Comparison of the reactivity of the Zn$^{2+}$ species verses the Mn$^{2+}$ by monitoring the hydrolysis of 4-nitrophenyl phosphate by UV–Vis spectroscopy shows after 24 h the Mn$^{2+}$ has hydrolysed three times more phosphate, indicating that the Mn$^{2+}$ is more active than the Zn$^{2+}$ complex.

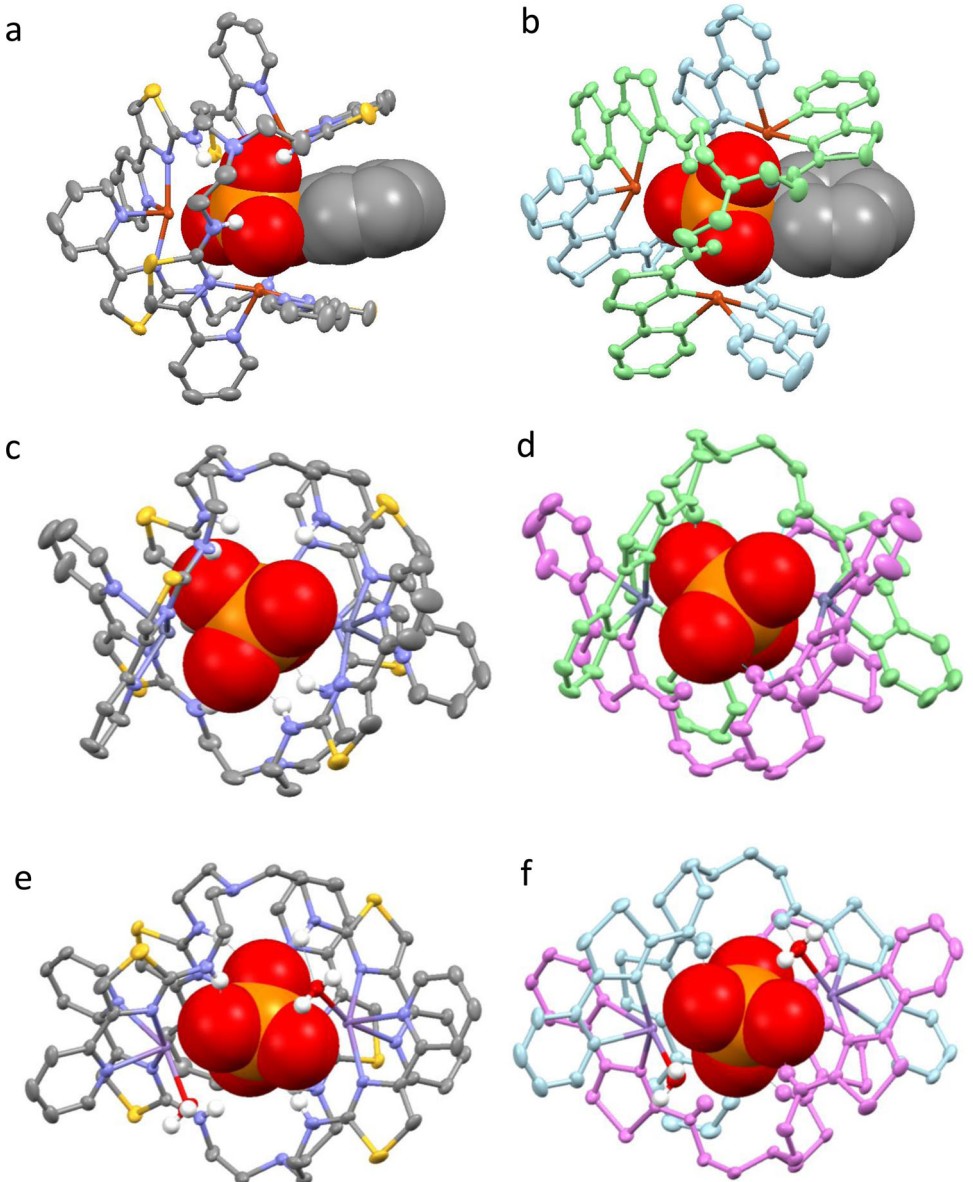

**Fig. 3 X-ray structural analysis of the reaction of [L₂Cu₃]⁶⁺, [L₂Zn₃]⁶⁺ and [L₂Mn₃]⁶⁺ with PhOPO₃²⁻. a** [L₂Cu₃(PhOPO₃)]⁴⁺. **b** [L₂Cu₃(PhOPO₃)]⁴⁺. **c** [L₂Zn₃(PO₄)]³⁺. **d** [L₂Zn₃(PO₄)]³⁺. **e** [L₂Mn₃ (H₂O)₂(PO₄)]³⁺. **f** [L₂Mn₃(H₂O)₂(PO₄)]³⁺. Colour code: dark orange, Cu(II); dark blue, Zn(II); purple, Mn; red, O; blue, N; yellow, S; grey, C; orange, P (apart from 2b, 2d and 2f where the ligand strands have been coloured for clarity).

**Activity of complexes towards cells**. The complexes [L₂Cu₃]⁶⁺ and [L₂Zn₃]⁶⁺ possess both potent and selective activity against most of the cancer cell lines tested compared to three non-cancer cell models utilised (Fig. 5a–d). For [L₂Zn₃]⁶⁺, IC₅₀ values towards the cancer cell lines ranged from 70 ± 13 nM against HCT116 p53⁻/⁻ and up to 59.07 ± 5.60 μM against MiaPaCa2. IC₅₀ values for both complexes were mostly sub-μM towards cancer cells (HT-29, DLD-1, HCT116 (p53 wild type and null), BxPC3, A549 and H460 cell lines) with the exceptions being the pancreatic cancer cell line PSN1 and the GBM1 glioblastoma cancer stem cell model[32,33] where IC₅₀ values were >1 μM. Cancer stem cells are typically chemoresistant[39], however, importantly both complexes showed preferential activity towards the GBM1 cells compared to all three non-cancer cell models which included adult human brain progenitor cells (NP1)[32,33]. The MiaPaCa2 pancreatic cancer cell line was however inherently resistant to both complexes with IC₅₀ values >10 μM (Fig. 5a, b).

The magnitude of selectivity towards cancer cells was marked and for both [L₂Cu₃]⁶⁺ and [L₂Zn₃]⁶⁺ was over 10-fold for most of the cancer cell lines compared to all three non-cancer cell models (Fig. 5c, d). Remarkably, for [L₂Zn₃]⁶⁺ selectivity indices of over 2000 were obtained in the case of HCT116 p53⁻/⁻ cancer cells compared to ARPE-19 and MCF10A non-cancer cells (Fig. 5d). For the [L₂Cu₃]⁶⁺ complex, selectivity indices for many of the cancer cell lines were >100 with the [L₂Zn₃]⁶⁺ complex resulting in even higher SIs. Interestingly, however, whereas activity of the [L₂Cu₃]⁶⁺ complex was very similar against the three non-cancer cell models, the NP1 brain progenitor cells[32,33] showed increased sensitivity to the Zn²⁺ complex.

Whilst the potency of both the Zn²⁺ and Cu²⁺ complexes (Fig. 5a, b) compares favourably with that of the clinically approved platinates (Fig. 5e), selectivity of the complexes is significantly superior under the same in vitro experimental conditions (Fig. 5c, d cf. 5f). Formation of the trinuclear species is

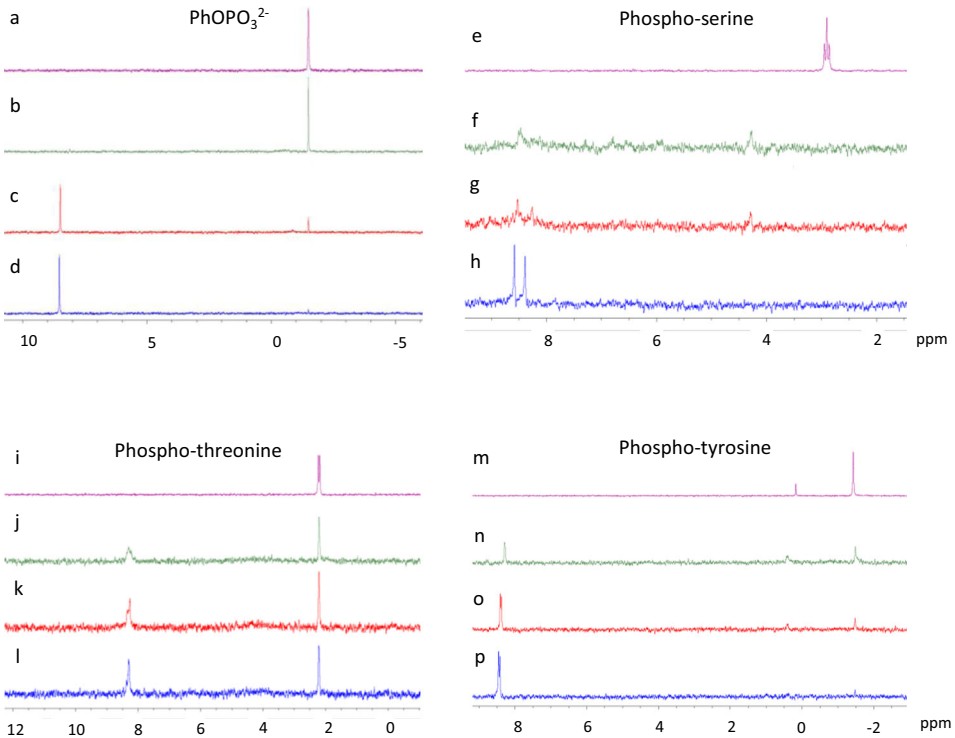

**Fig. 4 Phosphatase activity of [L₂Zn₃]⁶⁺.** ³¹P NMR spectra using different substrates including PhOPO₃²⁻ (spectra **a–d**), serine phosphate (spectra **e–h**), threonine phosphate (spectra **i–l**) and tyrosine phosphate (spectra **m–p**). Specific details for each ³¹P NMR spectra are as follows: Spectra **a**, **e**, **i** and **m** represent substrate alone (44 h incubated @ 37 °C); Spectra **b**, **f**, **j** and **n** represents [L₂Zn₃]⁶⁺ plus substrate (t = 0 min); Spectra **c**, **g**, **k** and **o** represents [L₂Zn₃]⁶⁺ plus substrate incubated @ 37 °C for 19 h; Spectra **d**, **h**, **l** and **p** represents [L₂Zn₃]⁶⁺ plus substrate incubated @ 37 °C for 44 h. The ³¹P NMR of [L₂Zn₃(PO₄)]³⁺ gives a signal at 8.6 ppm. The doubling up of some of the ³¹P signals in [L₂Zn₃(PO₄)]³⁺ (**h**, **p** and **l**) is attributed to formation of a mixture of diastereoisomers between the racemic cryptand and the resolved chiral amino acids which will form an ion-pair ([L₂Zn₃(PO₄)](RCH(NH₂)CO₂)²⁺) and does not occur with the achiral phenyl phosphate. ³¹P NMR spectra of the phosphate esters are ¹H-coupled (**a**, **e**, **i** and **m**) and the remainder are ¹H-decoupled.

important as solutions containing stoichiometric amounts of metal and ligand (e.g. favouring the formation of the mononuclear complex [LM]²⁺) lack selectivity in vitro as does the free ligand (Supplementary Fig. 9). In contrast, the Mn²⁺ complex, although a potent cytotoxin in vitro (Fig. 5e), showed only modest preferential selectivity (~2-fold) towards cancer cells which was comparable to that of the platinates (Fig. 5f).

The encapsulation of different specific anions (e.g. PO₄³⁻, SO₄²⁻ or PhOPO₃²⁻) into the [L₂Zn₃]⁶⁺ and [L₂Cu₃]⁶⁺ complexes at the point of self-assembly prior to any cell exposure impacts on both activity and selectivity. The effect is both anion and cell line dependent (Fig. 6 and see Supplementary Figs. 6–8) with the inclusion of different anions having either minimal effect or causing an increase or decrease in potency depending on the cell line. Against the PSN1 cell line for example, both [L₂Zn₃(SO₄)]⁴⁺ and [L₂Zn₃(O₃POPh)]⁴⁺ are significantly more active than [L₂Zn₃(PO₄)]³⁺ or [L₂Zn₃]⁶⁺ and this translates into improved selectivity indices using the ARPE-19 cell line as baseline (Fig. 6). A similar but smaller increase in potency was also observed against HCT116 p53⁺/⁺ (but not the p53 null variant) and H460 cells treated with [L₂Zn₃(SO₄)]⁴⁺ and [L₂Zn₃(O₃POPh)]⁴⁺ leading to a corresponding increase in relative selectivity. In contrast, the inclusion of anions reduced the activity of [L₂Zn₃]⁶⁺ against HT-29, DLD-1, BxPC3 and A549 cells resulting in a corresponding reduction in relative selectivity (Fig. 6). Similar results were obtained with [L₂Cu₃]⁶⁺ complexes with PO₄³⁻, SO₄²⁻ or PhOPO₃²⁻ anions (see Supplementary Figs. 6–8). It is noteworthy that [L₂Zn₃]⁶⁺ pre-encapsulated with PO₄³⁻ resulted in similar potencies, except against MCF10A cells, to [L₂Zn₃]⁶⁺ alone. Due to

significant quantities of phosphate in the cell culture media (~1 mM), it is likely that [L₂Zn₃]⁶⁺ would encapsulate PO₄³⁻ to form [L₂Zn₃(PO₄)]³⁺. Interestingly, however, the media used to culture MCF10A cells (Eagles MEM) contains more sulfate and lower amounts of phosphate than the culture media for most of the other cell lines and this may affect speciation and activity. In contrast, the pre-encapsulated sulfate and phenyl phosphate derivatives show some clear differences in potency and selectivity to [L₂Zn₃]⁶⁺ depending on the cell line. Studies using ³¹P NMR demonstrates that, whilst phosphate is present in the media, this anion does not entirely displace the pre-encapsulated anion and in a competitive experiment [L₂Zn₃(SO₄)]⁴⁺ is still the major species (see Supplementary Fig. 4.6). The mechanistic basis for these differential effects requires further investigation. However, these results demonstrate that the activity of these complexes can be readily modulated or potentially 'tuned' towards different cancer cells by altering the metal and/or the anion providing an inherently flexible platform for drug discovery.

**Mechanistic kinase studies.** Given the ability of the Zn²⁺ complex to dephosphorylate amino acids serine, tyrosine and threonine (Fig. 4) and the Cu²⁺ complex to bind phenyl phosphate (Fig. 3), this led us to examine whether the complexes affect the activity of kinases. This was assessed in a cell-free screen of 140 kinases using purified recombinant human kinases and substrate peptides incubated in the presence of ³³P ATP and 10 μM complex or solvent control. [L₂Zn₃]⁶⁺ and [L₂Cu₃]⁶⁺ inhibited the activity of multiple kinases to differing extents, with the most potently inhibited kinases being inhibited by near to 100% (Fig. 7a, b, Supplementary Fig. 10).

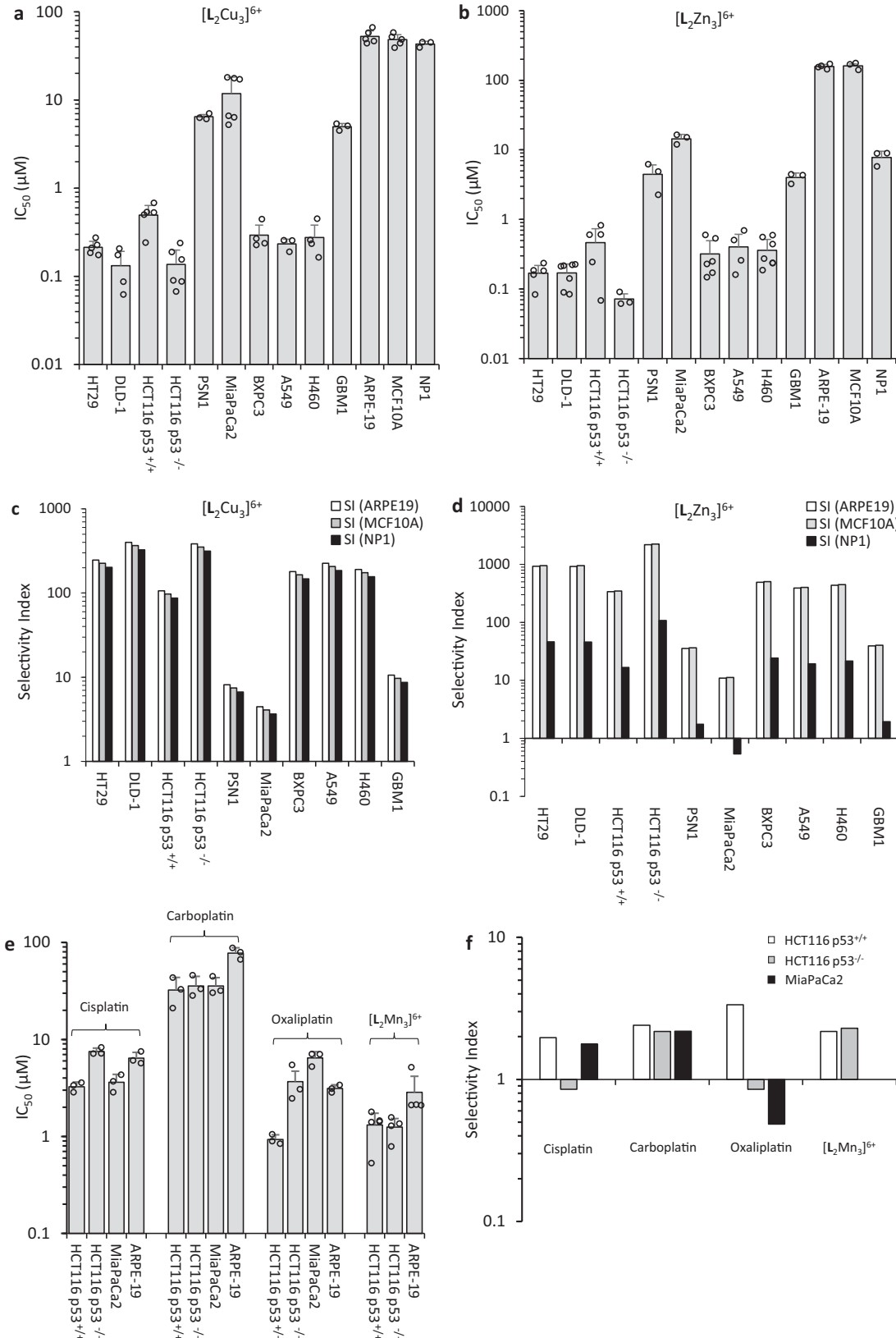

[$L_2Cu_3$]$^{6+}$ has inhibitory activity against a larger number of kinases than [$L_2Zn_3$]$^{6+}$ (Supplementary Fig. 10) which may reflect the fact that selectivity indices for [$L_2Cu_3$]$^{6+}$ are comparatively lower than those for [$L_2Zn_3$]$^{6+}$. In some cases, there is selective inhibition of

specific kinases by either [$L_2Zn_3$]$^{6+}$ or [$L_2Cu_3$]$^{6+}$ (Supplementary Fig. 10). For a few kinases, activity was increased by [$L_2Cu_3$]$^{6+}$ or [$L_2Zn_3$]$^{6+}$ (Fig. 7c, d), the most striking example being that of the proto-oncogene Src, a non-receptor tyrosine kinase[40]. Across the

**Fig. 5 Chemosensitivity response of a panel of human cancer and non-cancer cell lines to 96 h continuous exposure to self-assembling test compounds. a, b** The potency of compounds tested against cancer (HT-29, DLD-1, HCT116 $p53^{+/+}$ and $p53^{-/-}$, PSN1, MiaPaCa2, BxPC3, A549, H460 and GBM1) and non-cancer cells (ARPE-19, MCF10A and NP1). Each value represents the mean $IC_{50} \pm SD$ from a minimum of three independent experiments. The selectivity index (SI) for $[L_2Cu_3]^{6+}$ (**c**) and $[L_2Zn_3]^{6+}$ (**d**) for the indicated cancer cell lines; SI is defined as the mean $IC_{50}$ against the particular non-cancer cell line model divided by the mean $IC_{50}$ against the particular cancer cell line. SI values > 1 indicate that the test compound is more active against the particular cancer cell line than the corresponding non-cancer cells. As the SI value is calculated using the mean $IC_{50}$ values, experimental error is not included in these figures. **e** $IC_{50}$ values for the clinically approved platinates (cisplatin, oxaliplatin and carboplatin) and $[L_2Mn_3]^{6+}$ ($n =$ a minimum of three independent experiments and each value represents the mean $IC_{50} \pm SD$ from a minimum of three independent experiments) and **f** the corresponding SI results.

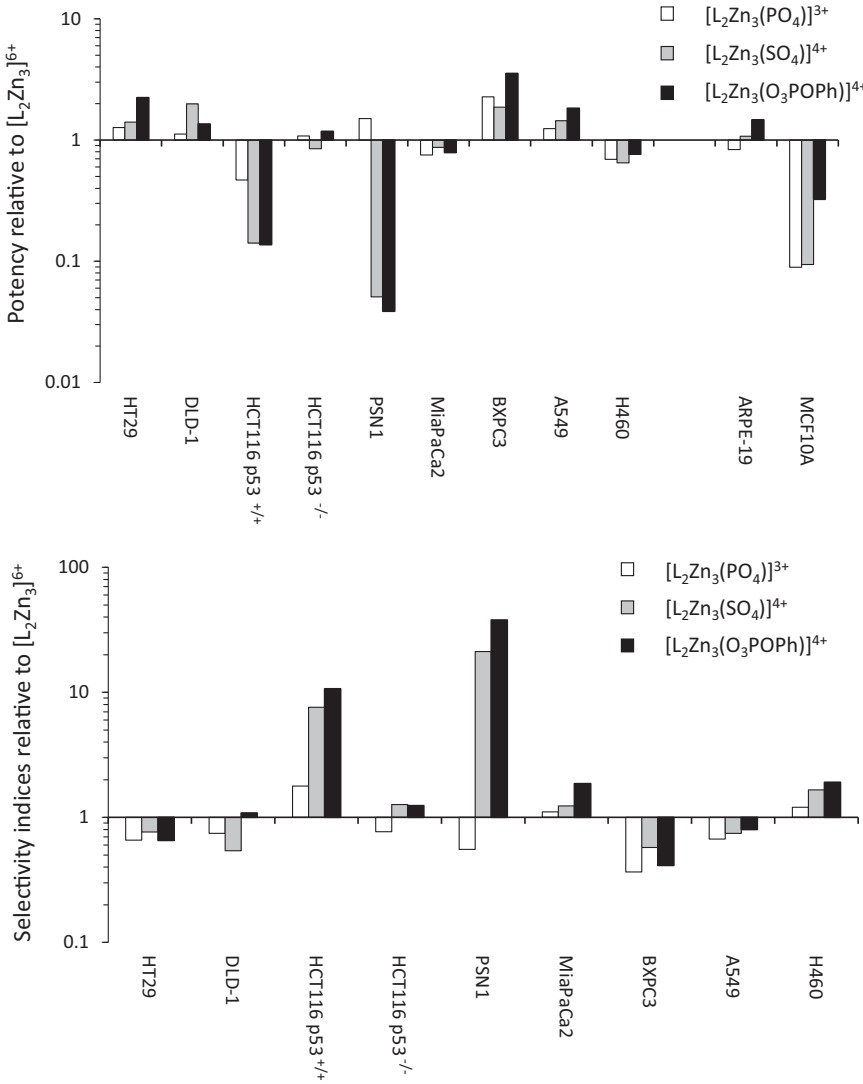

**Fig. 6 Effect of the complex anion on potency and selectivity. a** The effect of various anions on the potency of the $Zn^{2+}$ complex; this is expressed as relative potency which is defined as the $IC_{50}$ of $[\mathbf{L}_2Zn_3(PO_4)]^{3+}$, $[\mathbf{L}_2Zn_3 (SO_4)]^{4+}$ or $[\mathbf{L}_2Zn_3(O_3POPh)]^{4+}$ divided by the $IC_{50}$ of $[\mathbf{L}_2Zn_3]^{6+}$ (values <1 and >1 indicate increased and decreased potency respectively). **b** Effects of the anion on selectivity; these results are expressed as relative SI defined as the SI of $[\mathbf{L}_2Zn_3(PO_4)]^{3+}$, $[\mathbf{L}_2Zn_3(SO_4)]^{4+}$ or $[\mathbf{L}_2Zn_3(O_3POPh)]^{4+}$ divided by the SI of $[\mathbf{L}_2Zn_3]^{6+}$ (values >1 and <1 indicate increased and decreased selectivity respectively).

kinome, we observed inhibition of kinases in the TK and CAMK family for $[\mathbf{L}_2Zn_3]^{6+}$ and the CAMK and AGC family for $[\mathbf{L}_2Cu_3]^{6+}$ (Fig. 7e, f), however whether this reflects an overall preference of the compounds requires further investigation.

Kinases are a major cellular drug target in the oncology field with the majority of kinase inhibitors that are clinically approved being ATP competitive inhibitors[41]. Against human AMPKα as one of the most potently inhibited kinases in the screen, assays

showed both $[\mathbf{L}_2Cu_3]^{6+}$ and $[\mathbf{L}_2Zn_3]^{6+}$ complexes to be neither ATP competitive kinase inhibitors nor substrate competitive inhibitors (Fig. 7g, h) suggesting a non-competitive allosteric mechanism of kinase inhibition.

Other potential mechanisms were considered for the kinase screen inhibition 'read-out' including, i) direct ATP hydrolysis by the complex(es) and differing Km of the kinases for ATP, and, ii) dephosphorylation of the peptide substrate by the phosphatase

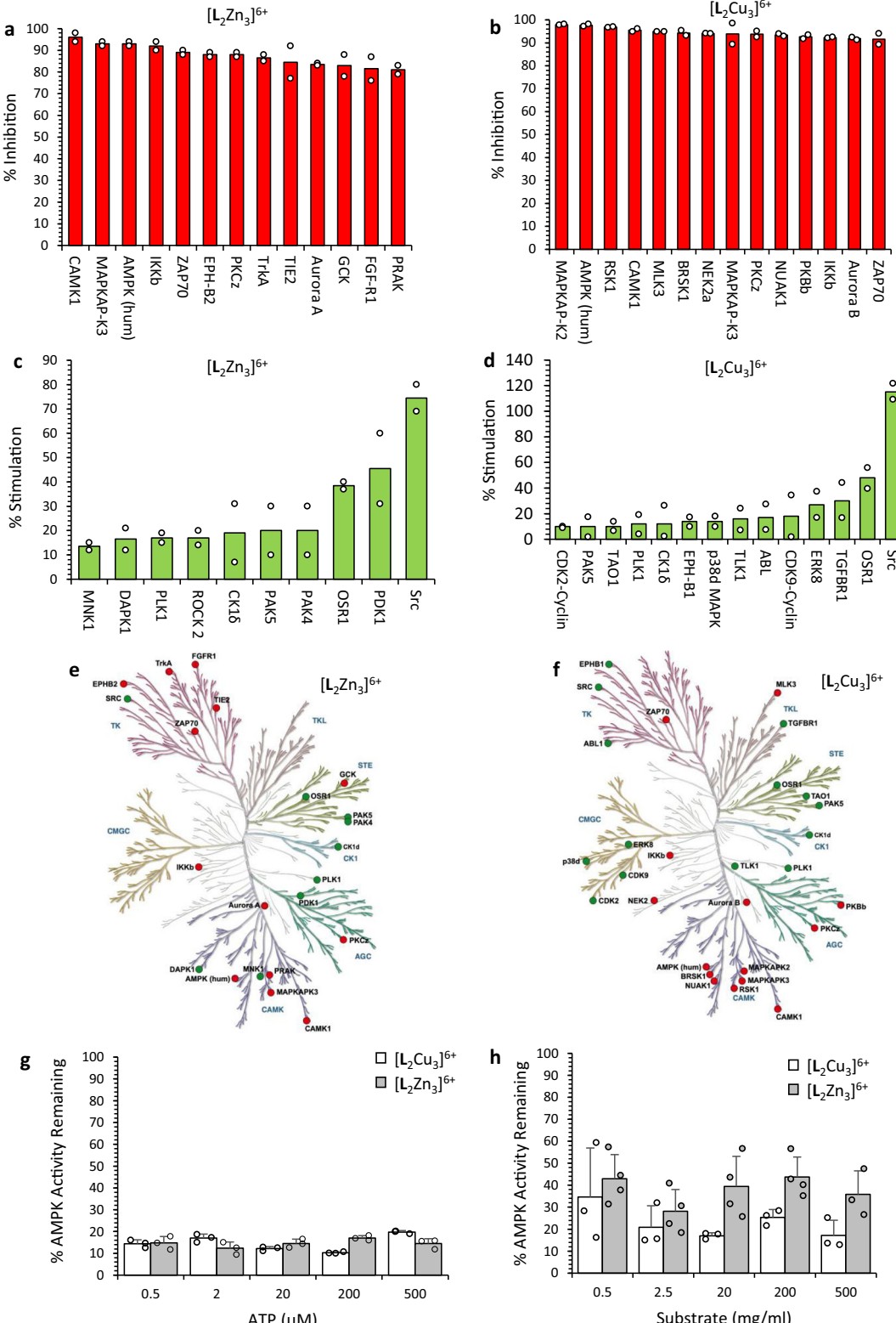

**Fig. 7 Effects of [L₂Zn₃]⁶⁺ and [L₂Cu₃]⁶⁺ on the activity of purified human kinases.** Percentage inhibition (red) of the indicated kinases by [L₂Zn₃]⁶⁺ (**a**) and [L₂Cu₃]⁶⁺ (**b**) respectively at a concentration of 10 μM (n = 2); full results are presented in the ESI. Kinases whose activity is stimulated (green) by [L₂Zn₃]⁶⁺ (**c**) or [L₂Cu₃]⁶⁺ (**d**) complexes. Kinome map showing kinases inhibited (red) or stimulated (green) by [L₂Zn₃]⁶⁺ (n = 2) (**e**) or [L₂Cu₃]⁶⁺ (**f**). Illustration reproduced courtesy of Cell Signalling Technology Inc (www.cellsignal.com). **g**, **h** AMPK kinase inhibition by [L₂Zn₃]⁶⁺ or [L₂Cu₃]⁶⁺ is non-competitive with ATP or substrate. **g** Influence of increasing ATP concentration (0.5–500 μM) on the inhibition of AMPK kinase activity by [L₂Zn₃]⁶⁺ or [L₂Cu₃]⁶⁺ with the percentage of remaining kinase activity at each tested ATP concentration indicated. **h** Influence of increasing substrate concentration (1–15 mg/ml) on the inhibition of AMPK kinase activity by [L₂Zn₃]⁶⁺ or [L₂Cu₃]⁶⁺ with the percentage of remaining kinase activity at each tested substrate concentration indicated. Results in **g** and **h** are presented as mean ± SD from a minimum of three independent biological repeats.

activity of the Zn$^{2+}$ complex resulting in cycles of peptide phosphorylation by active recombinant enzyme and dephosphorylation by the complex. However, mass spectroscopy showed little or no hydrolysis of ATP when incubated with Zn$^{2+}$ or Cu$^{2+}$ complex alone (Supplementary Fig. 2.30). Similarly, neither complex resulted in dephosphorylation of a purified phosphorylated AMPK peptide substrate (see Supplementary Fig. 4.3).

A plausible alternative explanation of the observed effects which can be reconciled with both selective kinase inhibition and activation is that that the Zn$^{2+}$ and Cu$^{2+}$ complexes are modulating key regulatory phospho-sites on the kinases themselves leading to enhanced or repressed kinase activity. In the case of the Zn$^{2+}$ complex, dephosphorylation of specific phosphorylated regulatory amino acids through its phosphatase activity (Fig. 4) may lead to either kinase inhibition or activation. For the Cu$^{2+}$ complex, it was hypothesised that it may bind to specific phosphorylated amino acids given its ability to bind but not hydrolyse phenyl phosphate (Fig. 3). This could affect kinase activity through influencing docking or binding of other proteins (e.g. SH2- domain-containing activating or inhibitory proteins)[42] or through steric or structural effects. One possible mechanism of kinase inhibition by either complex not formally investigated is the blocking of ADP release or it may be that multiple mechanisms contribute to the observed kinase inhibition or activation depending on the particular kinase.

To investigate these hypotheses further, regulatory phospho-amino acids of two key kinases identified by the kinase screen, AMPK and Src, were analysed following incubation of the recombinant kinases with Zn$^{2+}$ or Cu$^{2+}$ complexes. Phosphorylation of AMPKα at threonine 172 (T172P) stimulates AMPK activity[43] and treatment of AMPK with either $[L_2Zn_3]^{6+}$ or $[L_2Cu_3]^{6+}$ significantly reduced T172P levels detectable by immunoblotting at a molecular weight of ~63 kDa (Fig. 8a).

Decreased T172P is consistent with the inhibition of AMPK by both $[L_2Zn_3]^{6+}$ and $[L_2Cu_3]^{6+}$ observed in the kinase screen (Fig. 7a, b) but subtle differences in effects of the $[L_2Zn_3]^{6+}$ and $[L_2Cu_3]^{6+}$ are revealed. In the case of $[L_2Cu_3]^{6+}$, high molecular weight bands of >250 kDa are detected with the T172P specific AMPKα antibody suggesting that this compound is binding to the kinase, consistent with its ability to bind or encapsulate phosphoanionic molecules as demonstrated in Fig. 3. $[L_2Zn_3]^{6+}$ on the other hand causes a clear reduction in T172P and no high molecular weight bands consistent with phosphatase activity towards this phosphorylated amino acid of recombinant AMKPα (Fig. 4). These effects of $[L_2Zn_3]^{6+}$ are not restricted to T172P of AMPKα, with phosphorylated levels of S108 of the non-catalytic AMPKβ subunit also decreasing following incubation with $[L_2Zn_3]^{6+}$ in a time- and concentration- dependent manner (Supplementary Fig. 11). Phosphorylation of S108 of AMPKβ increases AMPK activity[44] indicating at least two potential mechanisms by which $[L_2Zn_3]^{6+}$ can inhibit AMPK kinase activity (Fig. 7a, g, h).

For Src, effects of the complexes on the key regulatory phospho-amino acids Y527 and Y416 were examined (Fig. 8a). Phosphorylation at Y527 in the Src C-terminal domain is known to decrease Src activity whereas autophosphorylation at Y416 in the activation loop of the kinase domain increases Src activity[40,45]. Here, incubation of recombinant Src with $[L_2Zn_3]^{6+}$ resulted in a ~30% decrease in Y527P relative to total Src levels and a ~5-fold increase in Y416P levels (Fig. 8a), both of which combined would be expected to result in enhanced Src kinase activity as was observed in the kinase screen (Fig. 7c, d). The decrease in Y527P is consistent with selective phosphatase activity of $[L_2Zn_3]^{6+}$. Y527 dephosphorylation reduces allosteric inhibition of Src kinase activity by the C-terminal domain, enabling autophosphorylation

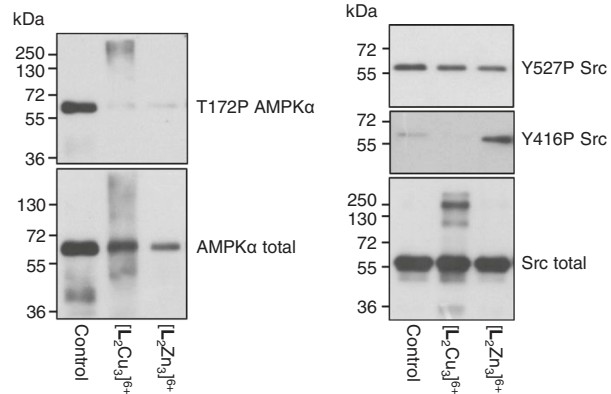

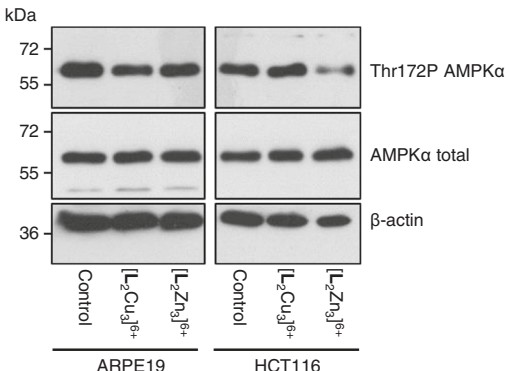

**Fig. 8 Western blot analysis of purified recombinant AMPK and Src following exposure to $[L_2Zn_3]^{6+}$ and $[L_2Cu_3]^{6+}$ and the effects of $[L_2Zn_3]^{6+}$ and $[L_2Cu_3]^{6+}$ treatment on cellular phospho-Thr172 AMPKα levels. a** Purified AMPK and Src enzymes were incubated with complexes (50 µM) for 4 h in the presence of ATP prior to analysis. **b** ARPE-19 non-cancer cells and HCT116 cancer cells were treated with 10 µM $[L_2Cu_3]^{6+}$ or $[L_2Zn_3]^{6+}$ for 4 h prior to harvesting and immunoblot analysis. Similar results were obtained in a minimum of $n = 2$ independent experiments.

of Src at Y416[40]. Similar to AMPK, incubation of $[L_2Cu_3]^{6+}$ with Src resulted in higher molecular weight bands detected with the total Src antibody which were not observed with control or $[L_2Zn_3]^{6+}$ treatments, suggesting binding of $[L_2Cu_3]^{6+}$ to Src. Y416P could not be detected raising the possibility that $[L_2Cu_3]^{6+}$ binding to Src masks detection of this epitope, however, further studies are required to unravel the detailed mechanism as to how $[L_2Cu_3]^{6+}$ binding to Src may increase its activity.

**Intracellular $[L_2Zn_3]^{6+}$ and decreased cellular phospho-Thr172 AMPKα.** Many of the kinases inhibited or activated in the cell-free kinase screen are intracellular raising the important question as to whether $[L_2Zn_3]^{6+}$ complexes are actually able to enter cells to potentially access these kinases or whether they exert their selective anti-cancer effects outside of the cell or at the cell membrane. The cell permeable Zn-binding dye TSQ[46] was used as a surrogate marker to assess presence of $[L_2Zn_3]^{6+}$ within cells and whether there were any differences in levels or localisation between the chemoresistant ARPE-19 non-cancer and the chemosensitive HCT116 p53$^{+/+}$ cancer cell line. The results indicate that the complex is taken up by both cell lines suggesting that differential chemosensitivity is unlikely to be caused by differential drug uptake (Supplementary Fig. 16). It also suggests that $[L_2Zn_3]^{6+}$ is able to directly access and potentially interact with,

or dephosphorylate, at least some of the intracellular kinases that it inhibits or activates in the cell-free kinase screen. We cannot, however, currently exclude the possibility that anti-cancer activity of the complexes may be exerted without the complexes entering the cells, for example by the complexes transiently interacting with kinases located at the cell membrane (e.g. receptor tyrosine kinases) affecting downstream phospho-signalling. Analyses of cellular phospho-Thr172 AMPKα levels following 4 h treatment with 10 μM $[\mathbf{L}_2\mathbf{Cu}_3]^{6+}$ or $[\mathbf{L}_2\mathbf{Zn}_3]^{6+}$ revealed decreased T172P AMPKα selectively in the HCT116 cancer cells with $[\mathbf{L}_2\mathbf{Zn}_3]^{6+}$ treatment (Fig. 8b).

**Selective induction of autophagy and cancer cell ATP depletion.** Given the kinase screen results indicating that the $[\mathbf{L}_2\mathbf{Cu}_3]^{6+}$ and $[\mathbf{L}_2\mathbf{Zn}_3]^{6+}$ complexes can inhibit multiple kinases as well as activating several others (Fig. 7 and see Supplementary Fig. 10) and the common dysregulation of phospho-signalling in cancers[38,47], this provides a plausible mechanism by which they might exert their cancer selective activity. Phenotypically, by 40 h $[\mathbf{L}_2\mathbf{Zn}_3]^{6+}$ induced the appearance of vacuoles in the HCT116 p53$^{+/+}$ and p53$^{-/-}$ cancer cells which were shown to be autophagic using the established autophagic tracer dye CYTO-ID[48]. The Mn$^{2+}$ complex also induced autophagy whereas $[\mathbf{L}_2\mathbf{Cu}_3]^{6+}$ did not (Fig. 9a and see Supplementary Figs. 13–15).

Autophagy is a catabolic process that is induced in response to metabolic stresses including low ATP levels and starvation[43]. We hypothesised that the induction of autophagy by Zn$^{2+}$ and Mn$^{2+}$ complexes could be due to cellular ATP depletion resulting from their protein phosphatase activity and repeated futile cycles between protein phosphorylation by constitutively active oncogenic kinases[47] and dephosphorylation by the complexes. In support of this hypothesis, Zn$^{2+}$ and Mn$^{2+}$ complexes both caused a dose-dependent decrease in ATP levels (Fig. 9b and see Supplementary Fig. 17). This was much more pronounced in the HCT116 cancer cells than in ARPE-19 non-cancerous cells with $[\mathbf{L}_2\mathbf{Zn}_3]^{6+}$, indicating cancer selective ATP depletion correlating with the excellent cancer cell selectivity indices of the Zn$^{2+}$ complex (Fig. 5d). There was less difference in the magnitude of ATP depletion between the HCT116 cancer cells and the ARPE-19 non-cancer cells with Mn$^{2+}$ complex treatment (see Supplementary Fig. 17), consistent with its more modest selectivity indices (Fig. 5f). This may relate to more promiscuous phosphatase activity of the Mn$^{2+}$ complex or its higher rates of hydrolysis (see Supplementary Fig. 5.1.). Another possibility, however, for the observed ATP depletion (Fig. 9b) is decreased ATP generation through the inhibition of glycolysis and/or mitochondrial respiration by the complexes. 1 h treatment with 25 μM $[\mathbf{L}_2\mathbf{Zn}_3]^{6+}$ caused a modest selective decrease in glycolysis in the HCT116 cancer cells whilst the more metabolically quiescent ARPE-19 cells were unaffected (Fig. 9c). 20 h treatment substantially impaired both mitochondrial respiration and glycolysis in the HCT116 cancer cells (Fig. 9d), likely contributing to the decrease in ATP levels.

It is currently unclear whether the effects of $[\mathbf{L}_2\mathbf{Zn}_3]^{6+}$ on glycolysis and mitochondrial respiration are linked to its phosphatase/kinase inhibitory activity or are via an independent mechanism. A number of key metabolic enzymes are kinases themselves (e.g. hexokinase, pyruvate kinase, phosphoglycerate kinase)[49] whilst others are regulated by phosphorylation (e.g. pyruvate dehydrogenase, lactate dehydrogenase A (LDH-A)). $[\mathbf{L}_2\mathbf{Zn}_3]^{6+}$ was found to modestly decrease p-Y10 LDH-A (phosphorylation of Y10 increases LDH-A activity) in ARPE-19 cells and to a lesser extent in HCT116 cells (see Supplementary Fig. 18).

Immunoblot analyses suggest that the observed autophagy (Fig. 9a) is a compensatory catabolic response to sustain ATP

levels and prevent bioenergetic failure and death with 40 h 5 μM $[\mathbf{L}_2\mathbf{Zn}_3]^{6+}$ inducing activation of the 'low ATP' sensing kinase AMPK[43] in HCT116 cancer cells but not in ARPE-19 non-cancer cells (Fig. 10).

Thus, cellular levels of T172P of AMPKα were increased relative to total AMPKα levels specifically in the HCT116 cancer cells by $[\mathbf{L}_2\mathbf{Zn}_3]^{6+}$ but not by $[\mathbf{L}_2\mathbf{Cu}_3]^{6+}$ (Fig. 10). T172 AMPKα phosphorylation is induced by low cellular ATP levels and increased AMP or ADP levels, enabling competitive binding of AMP/ADP to the γ regulatory subunit of AMPK enabling AMPKα phosphorylation by one of the AMPK upstream kinases[43]. Thus, paradoxically, $[\mathbf{L}_2\mathbf{Zn}_3]^{6+}$ causes dephosphorylation of T172 AMPKα and kinase inhibition in a cell-free system (Fig. 7a, g, h) and in the HCT116 cancer cells with brief exposure (Fig. 8b) consistent with its phosphatase activity (Fig. 4) but longer treatment results in increased cellular levels of T172P AMPK (Fig. 10) which is attributed to selective ATP depletion (Fig. 9b).

Following $[\mathbf{L}_2\mathbf{Zn}_3]^{6+}$ treatment of HCT116 cancer cells, a small decrease (~10%) in total levels of Tyr- phosphorylated proteins was detected by immunoblotting using a pan phospho-tyrosine antibody although there was also evidence of increased p-Y of some proteins (*) (Fig. 10). Importantly, both $[\mathbf{L}_2\mathbf{Cu}_3]^{6+}$ and $[\mathbf{L}_2\mathbf{Zn}_3]^{6+}$ treatment resulted in increased levels of the p53 tumour suppressor protein which is induced by many different types of cellular stress resulting in the co-ordination of an appropriate stress response which can include cell cycle arrest or cell death induction[50]. Whilst p53 levels were increased by ~1.7 fold by $[\mathbf{L}_2\mathbf{Cu}_3]^{6+}$ and ~2.7 fold by $[\mathbf{L}_2\mathbf{Zn}_3]^{6+}$ treatment in the HCT116 cancer cells, levels of p53 actually decreased with treatment in the ARPE19 non-cancer cells further indicating differential effects of the complexes on cancer versus non-cancer cells (Fig. 10). The earlier chemosensitivity results indicate, however, that the cytotoxicity of the $[\mathbf{L}_2\mathbf{Cu}_3]^{6+}$ and $[\mathbf{L}_2\mathbf{Zn}_3]^{6+}$ is not dependent on p53; indeed both complexes were more active towards the HCT116 p53$^{-/-}$ cells than the isogenic p53$^{+/+}$ cells (Fig. 5a).

In summary, in this study three different metal-containing, self-assembled anion binding complexes are characterised and shown to have distinctive chemical and biological properties. Depending on the metal, the reactivity of the complexes towards different anionic species varied with the Zn$^{2+}$ and Mn$^{2+}$ complexes both showing significant phosphatase activity but with different rates of hydrolysis (Mn$^{2+}$ >> Zn$^{2+}$) whereas the Cu$^{2+}$ complex bound to, rather than hydrolysed, phospho-containing species. Significant selective activity towards particular cancer cells compared to non-cancer cells was shown by the Zn$^{2+}$ and Cu$^{2+}$ complexes by different mechanisms with evidence of modulation of multiple kinases via either binding (Cu$^{2+}$) or by de-phosphorylation (Zn$^{2+}$) of regulatory sites on kinases. Using an unbiased phospho-proteomics approach, future studies will aim to identify which phospho-sites on proteins are selectively dephosphorylated by $[\mathbf{L}_2\mathbf{Zn}_3]^{6+}$. This will include whether $[\mathbf{L}_2\mathbf{Zn}_3]^{6+}$ shows any activity towards phospho-serine/threonine proline motifs, a common regulatory motif in many signalling proteins. However, we cannot exclude the possibility of additional mechanisms contributing to the selective anti-cancer activity of $[\mathbf{L}_2\mathbf{Cu}_3]^{6+}$ and $[\mathbf{L}_2\mathbf{Zn}_3]^{6+}$. Zn$^{2+}$ and Mn$^{2+}$ complexes both induced cancer cell autophagy consistent with cellular ATP deficiency and bioenergetic failure. Further modulation of activity and selectivity profile by incorporation of different anions (e.g. PO$_4^{3-}$, SO$_4^{2-}$ or PhOPO$_3^{2-}$) pre-cell exposure indicates the ease of generating numerous 'modular' combinations of metal/anion binding self-assembling complexes that can differ in potency, selectivity and mechanism(s) of action towards disease.

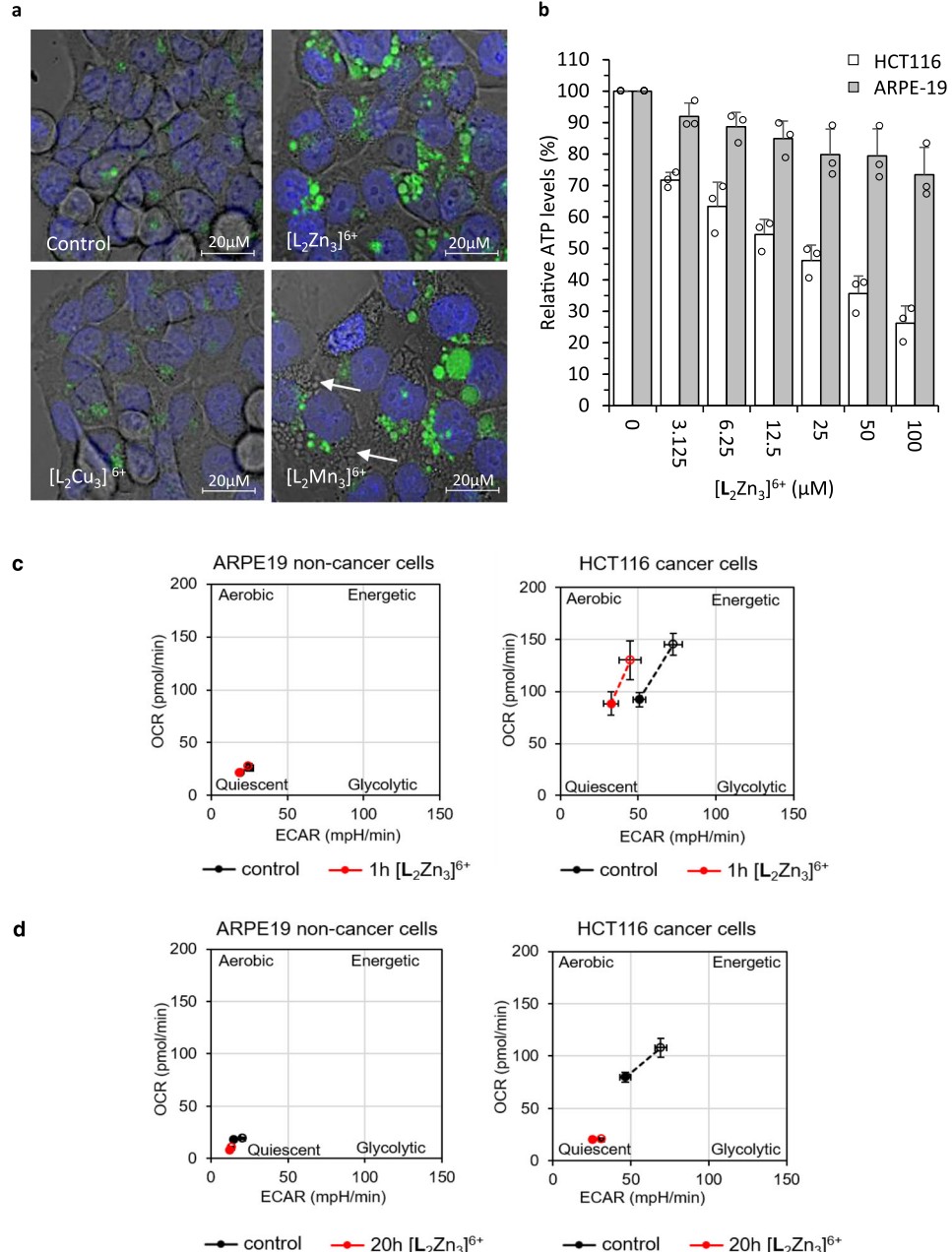

**Fig. 9 Effects of [L₂Zn₃]⁶⁺, [L₂Cu₃]⁶⁺ and [L₂Mn₃]⁶⁺ complexes on autophagy and of [L₂Zn₃]⁶⁺ on cellular ATP levels, glycolysis and mitochondrial respiration. a** Representative fluorescent images showing CYTO-ID autophagic staining (green) and Hoechst (blue) with bright field overlay of HCT116 p53⁻/⁻ cells treated with 3.125 μM [L₂Zn₃]⁶⁺, [L₂Cu₃]⁶⁺ or [L₂Mn₃]⁶⁺ for 40 h. White arrows indicate intracellular vacuoles. Similar results were obtained in $n = 3$ independent experiments. **b**, Cellular ATP levels in ARPE-19 non-cancer cells and HCT116 p53⁺/⁺ cancer cells with 20 h exposure to [L₂Zn₃]⁶⁺ at the indicated concentrations; ATP levels expressed relative to levels in vehicle control-treated cells. $n = 3$ biologically independent experiments. Mean ATP levels ± SD. **c**, **d** Metabolic phenograms showing cellular rate of oxygen consumption (OCR) as measure of mitochondrial respiratory rate and extracellular acidification rate (ECAR) as a measure of glycolytic rate in ARPE-19 and HCT116 cells (closed circles, basal metabolic rates). **c** Effect of 1 h 25 μM [L₂Zn₃]⁶⁺, **d** Effect of 20 h 25 μM [L₂Zn₃]⁶⁺. Open circles represent maximal ECAR and OCR (metabolic capacity/reserve) of control and [L₂Zn₃]⁶⁺ treated cells, determined by addition of a metabolic stress mixture (oligomycin and FCCP). Black line/circle indicates vehicle control-treated cells; red line/circle indicates [L₂Zn₃]⁶⁺ treated cells. $n = 3$ independent experiments; mean ± SD.

## Methods

**Synthesis**. Unless otherwise stated, all solvents and materials were purchased from either Sigma Aldrich, Fisher Scientific or Fluorochem and were used without further purification. ¹H, ¹³C, DEPT-135 and DEPT-90 NMR data was recorded on either a Bruker Fourier 300 MHz or Bruker Avance III (AVIII) 400 MHz spectrometer or a Bruker Avance Neo 600 MHz spectrometer. Mass spectra were obtained on an Agilent 6210 TOF MS with electrospray ionisation operating in positive ion mode and mass spectra of metal complexes were obtained on a Bruker Micro TOF-q LC mass spectrometer with electrospray ionisation operating in positive ion mode. Phosphorylated 'SAMS' peptide

HMRSAMS*GLHLVKRR (*phosphorylated on the serine residue) was obtained from PeptideSynthetics, Peptide Protein Research Ltd (>95% purity). Elemental analysis was performed on an Exeter Analytical CE440 Elemental Analyser. CAUTION: perchlorate anions are potentially explosive and should be treated with appropriate care. The complexes described were prepared and isolated only in small amounts (5–10 mg).

**Preparation of ligand L and its complexes**. The synthetic route is described by Supplementary Fig. 1.1. The ligand was prepared by a modification of a literature

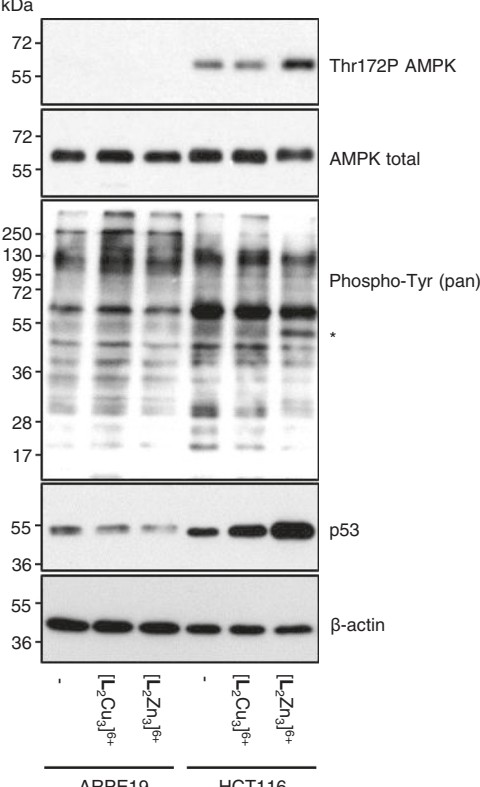

**Fig. 10 Immunoblots showing the differential effects of $[L_2Cu_3]^{6+}$ and $[L_2Zn_3]^{6+}$ treatment of HCT116 cancer cells and ARPE-19 non-cancer cells on key cellular proteins associated with cellular and metabolic stress.** Representative immunoblot images of the indicated proteins following cell exposure to 5 µM $[L_2Cu_3]^{6+}$ or $[L_2Zn_3]^{6+}$ for 40 h. Tyrosine phosphorylated proteins are indicated using a pan- phospho-Tyr antibody; β-actin as a loading control. Similar results were obtained in a minimum of $n = 2$ independent experiments.

procedure and the analytical data was identical to that reported[31]. For the chemical studies the complexes were prepared by direct reaction of metal perchlorate or tetrafluoroborate salts in MeCN (or acetone) and $H_2O$ with the ligand. The anion host species were prepared by addition of either $Na_2O_3POPh$, $Bu_4NH_2PO_4$, $Bu_4NHSO_4$ to solutions of the complex. The purity and composition of the complexes was confirmed by ESI-MS, $^1H/^{31}P$ NMR (where applicable) and elemental analysis. For the cytotoxicity studies all the complexes were prepared from the metal acetates[31].

**X-ray structural determination.** Single crystals of the compounds analysed via this method were grown from slow evaporation of a solution of MeCN/$H_2O$ (apart from $[LZn](ClO_4)_2$ where a mixture of acetone/MeCN was used). X-ray diffraction data was collected at 150(2) K on a Bruker D8 Venture diffractometer equipped with a graphite monochromated Mo(Kα) radiation source and a cold stream of $N_2$ gas. Solutions were generated by conventional heavy atom Patterson or direct methods and refined by full-matrix least squares on all $F^2$ data, using SHELXS-97 (for solution), SHELXL-2014 (for refinement) and SHELXL (for molecular graphics) software respectively[51]. In some examples Olex2 was used in refinement[52]. Absorption corrections were applied based on multiple and symmetry-equivalent measurements using SADABS[53]. Almost all the structures contained some form of disorder either with solvent molecules and/or counter anions (generally substitutional or rotation disorder). In these cases, the atoms were modelled using the *PART* instruction in the least squares refinement and refined over two positions. The anisotropic displacement parameters were treated with *SIMU*, *DELU* and in some cases *ISOR* where needed. Due to the diffuse nature of the electron density map the hydrogen atoms were not added to disordered solvent molecules. The structure $[L_2Zn_3(SO_4)](BF_4)_{3.5}$ contained extensively disordered tetrafluoroborate counter anions, one of which refined poorly and was modelled with 50% occupancy.

**NMR studies.** For $^1H$ NMR studies, reaction of $[L_2Zn_3]^{6+}$ with either $Na_2O_3POPh$ and $Bu_4NH_2PO_4$ was carried out in 10% $D_2O$ in $CD_3CN$. The NMR solutions were

prepared by dissolving the ligand **L** and $Zn(C_2H_3O_2)_2$ (in a ratio of 1:1.5) in $CD_3CN$ (600 µL). The relevant anion (0.5 equivalents w.r.t ligand) was dissolved in $D_2O$ (60 µL) and these solutions were then combined. Heating of the sample was carried out @ 80 °C in a water bath for the specified amount of time. For the $^{31}P$ NMR studies solutions were prepared by dissolving the ligand **L** and $Zn(C_2H_3O_2)_2$ (in a ratio of 1:1.5) in $d_6$-DMSO (200 µL). The anion (0.5 equivalents w.r.t ligand) was dissolved in HEPES (600 µL, 60 mmol @ $p$H 7.4) and these solutions were then combined. Where required heating of the sample was carried out by incubating the sample @ 37 °C and/or a further 2 h @ 80 °C in a water bath.

**UV–Vis studies.** UV–Vis studies were carried out on a Cary 60 UV–Vis spectrophotometer with Thermo cycler attachment and the wavelength range of 800–250 nm was scanned every hour for 43 h whilst the solution was heated at a constant temperature of 37 °C. Progress of hydrolysation of 4-nitrophenyl phosphate to 4-nitrophenol was measured by monitoring the increase in absorption at the wavelength of 400 nm (typical for 4-nitrophenol). HEPES buffer was prepared at $p$H 7.4. Solutions were prepared by dissolving the ligand **L** (3.1 mg, 0.005 mmol) and the relevant metal salt (1.5 equivalents with respect to ligand) in DMSO (5 mL). 4-nitrophenyl phosphate (18.6 mg, 0.050 mmol) was dissolved in HEPES buffer (~10 mL, 60 mmol). These solutions were then combined and made up volumetrically to 50 mL with HEPES buffer. Where required $Bu_4NH_2PO_4$ (0.5 equivalents with respect to ligand) was added to the HEPES buffer.

**Cell culture and chemosensitivity studies.** All cell lines were purchased from and maintained as monolayers as described by the American Type Culture Collection with the exception of the p53$^{+/+}$ and p53$^{-/-}$ isogenic clones of HCT116 human colorectal adenocarcinoma cells which were a kind gift from Professor Bert Vogelstein[54]. GBM1 cells were established from a primary GBM tumour by Heiko Wurdak and colleagues[32,33]. Non-cancerous adult brain neural progenitor (NP1) cells were derived from a patient undergoing surgery to treat epilepsy[32,33]. Cell culture and growth media for NP1 and GBM1 cells were as described[32,33]. The response of cells following a continuous 96-h exposure to test compounds and platinates was determined using the MTT assay. Potency was recorded as the IC$_{50}$ and the selectivity index (SI) was defined as the ratio of IC$_{50}$ values for non-cancer to cancer cell lines with values >1 representing selectivity for cancer cells as opposed to non-cancer cells[19].

**Cell line origins and culture media.** All cell lines used were maintained at low passage in antibiotic-free media. HT29, DLD-1, HCT116 p53$^{+/+}$ and HCT116 p53$^{-/-}$ are all colorectal adenocarcinoma cell lines derived from different individuals and harbour different combinations of oncogenic lesions (except for the HCT116 isogenic cancer cell clones that are genetically identical except for p53 status). PSN-1, BxPC-3 and MiaPaCa2 are pancreatic carcinoma cell lines and A549 and H460 are lung carcinoma cell lines. HT29, DLD-1, PSN-1, BxPC-3, A549 and H460 cell lines were cultured in RPMI-1640 growth media (Sigma) containing 2 mM L-glutamine, 1 mM sodium pyruvate and 10% foetal bovine serum (FBS). HCT116 (p53$^{+/+}$ and p53$^{-/-}$) and MiaPaCa2 cell lines were cultured in Dulbecco's Modified Eagle's Medium (Sigma), 2 mM L-glutamine and 10% FBS. The ARPE-19 retinal epithelial non-cancer cell line was cultured in DMEM/F12 media (Gibco), 2 mM L-glutamine, 1 mM sodium pyruvate and 10% FBS. The MCF10A non-cancerous human breast epithelial cell line was cultured in Minimal Essential Media Eagle (Sigma), 2 mM L-glutamine, 1 mM sodium pyruvate, 10% FBS and 1x non-essential amino acids (NEAA). NP1 and GBM1 cells were cultured on plasticware coated with poly-L-ornithine (5 µg/ml) and laminin (2.4 µg/ml)[32,33]. NP1 cells were grown in DMEM/F12 media (Gibco) supplemented with 5% FBS, 20 ng/ml hFGF, 20 ng/ml rhEGF, 0.5x B-27 supplement (Gibco), 0.5x N-2 supplement (Gibco) and 1x GLUTAMAX (Gibco). GBM1 cells were cultured in Neurobasal media (Gibco) supplemented with 40 ng/ml hFGF, 40 ng/ml rhEGF, 0.5x B-27 supplement (Gibco) and 0.5x N-2 supplement (Gibco).

**Cell seeding for chemosensitivity studies.** $[L_2(Metal)_3]^{6+}$ and all the other self-assembling complexes were freshly formed by adding DMSO to individual components and mixing together by pipetting. These were then further diluted in cell culture media such that the final DMSO concentration that cells were exposed to was 0.2% (vehicle control). Cisplatin, oxaliplatin and carboplatin were dissolved in phosphate buffered saline. Cell lines were seeded into 96 well plates at $2 \times 10^3$ cells per well and incubated overnight at 37 °C. GBM1 cancer stem-like cells were seeded at $3 \times 10^3$ cells per well and NP1 neural progenitors were seeded at $1.5 \times 10^3$ cells per well. The following day, media was removed and replaced with fresh media containing test compounds at a range of concentrations. Cells were incubated with test compounds for a further 96 h after which the media was removed and replaced with fresh media (200 µl/well). MTT was added (20 µl at 5 mg/ml) and cells were incubated for a further 4 h. Media and MTT were removed and formazan crystals were dissolved in 150 µl of DMSO and the absorbance of the resulting solution determined at 540 nm. Dose response curves were constructed, and the concentration required to reduce cell growth by 50% (IC$_{50}$) determined.

**Kinase inhibition screen.** $[L_2Cu_3]^{6+}$ and $[L_2Zn_3]^{6+}$ were submitted to the MRC Protein Phosphorylation and Ubiquitination Unit International Centre for Kinase Profiling (University of Dundee) and tested in duplicate (at 10 µM) against a series

of 140 purified human kinases in a cell-free kinase assay using $^{33}$P ATP[55,56]. The kinase gene IDs from the screen were then translated into common IDs (HGNC symbol) using biomaRt package[57,58] in R[59] and R Studio[60] and used as input to the KinMap programme[61]. The KinMap output was then manually annotated to include the top activated and inhibited kinases as shown in Fig. 7.

**ATP and substrate competition assays**. ATP competition kinase assays were performed by the MRC Protein Phosphorylation and Ubiquitination Unit International Centre for Kinase Profiling (University of Dundee). Inhibition of recombinant human AMPK (α1β2γ1) by 10μM [$L_2Cu_3$]$^{6+}$ or [$L_2Zn_3$]$^{6+}$ was evaluated at five different ATP concentrations (0.5, 2, 20, 200 and 500 μM ATP). Substrate competition assays were performed using the ADP-Glo Kinase Assay (Promega) and recombinant human AMPK (α1β2γ1) (Promega) with concentrations of the provided AMPK substrate (SAMStide substrate, Promega) ranging from 1 to 15 mg/ml. Following pre-incubation of substrate with kinase for 5 min 10 μM [$L_2Cu_3$]$^{6+}$, [$L_2Zn_3$]$^{6+}$ or solvent control and 50 μM ATP was added to start the kinase reaction. For determination of the effects of AMP on AMPK inhibition by [$L_2Zn_3$]$^{6+}$, the ADP-Glo Kinase Assay (Promega) was performed in presence of varying of AMP (0, 80, 160 μM).

**In vitro phosphatase assay with recombinant kinases**. 800 ng of purified recombinant human AMPK (α1β2γ1) or Src kinase (MRC PPU, International Centre for Kinase Profiling, University of Dundee) was incubated with 50 μM [$L_2Cu_3$]$^{6+}$, [$L_2Zn_3$]$^{6+}$, or vehicle control in 50 mM Tris HCl pH 7.5 containing 50 μM ATP for 4 h at 37 °C. For [$L_2Zn_3$]$^{6+}$ time course and concentration titration experiments, 800 ng recombinant human AMPK was used (α1β2γ1 AMPK isoform for pT172 AMPKα1 detection or α2β1γ1 AMPK isoform for pS108 AMPKβ1 detection). For evaluation of ATP dependency and influence of AMP, 50 μM ATP was replaced with 50 μM AMP or omitted. Reactions were stopped by addition of Laemmli's buffer (Biorad) to 1x and heating at 95 °C for 5 min before analysis by SDS PAGE and immunoblotting for detection of specific phosphorylated amino acids of AMPK and Src proteins.

**Metabolic studies**. ARPE-19 and HCT116 cells were seeded in 8-well XFp plates (Agilent) 48 h prior to metabolic flux analyses using the Agilent XFp extracellular flux analyser. Cell seeding densities were 2500 cells/well for ARPE-19 cells and 7500 cells/well for HCT116 cells in order to achieve equivalent cell confluency (~70% confluency and in log phase growth) at the time of analyses. Cells were treated with 25 μM of [$L_2Zn_3$]$^{6+}$ or solvent control (0.2% DMSO) for 1 h or 20 h before metabolic flux analyses using the Cell Energy Phenotype Assay (Agilent) in accordance with the manufacturer's instructions. For determination of maximal respiratory and glycolytic rates (metabolic reserve) following real-time quantification of basal respiratory and glycolytic rates, a 'metabolic stress mix' of 1 μM oligomycin and FCCP (0.5 μM for ARPE-19, 0.25 μM for HCT116) was injected into the media.

**Confocal microscopy imaging of autophagic cells**. Cells were seeded in Lumox chamber slides (8-well, Sarstedt) at a density of $1.25 \times 10^4$ cells/well (ARPE-19) or $5.625 \times 10^4$ cells/well (HCT116 p53$^{-/-}$ and HCT116 p53$^{+/+}$). 24 h after seeding, cells were treated with 3.125 μM of [$L_2Cu_3$]$^{6+}$, [$L_2Zn_3$]$^{6+}$, [$L_2Mn_3$]$^{6+}$, or vehicle control for 40 h before removal and addition of CYTO-ID Green autophagic dye (1:500 dilution, Enzo Life Sciences) and Hoechst 3342 nuclear counter stain for 30 min at 37 °C. Media was then replaced before imaging using a Zeiss LSM880 confocal laser scanning microscope.

**TSQ cellular localisation studies**. 48 h following cell seeding in Lumox chamber slides (8-well, Sarstedt), cells (ARPE-19 and HCT116 p53$^{+/+}$ were treated for 1 h at 37 °C with 25 μM of [$L_2Zn_3$]$^{6+}$ or DMSO solvent control before removal and washout three times using complete phenol-red free media. 25 μM TSQ (Enzo Life Sciences) was then added for 30 min for detection of cellular zinc before its washout (three washes) and analyses by confocal microscopy (Zeiss LSM880).

**Quantification of cellular ATP levels**. ARPE-19 and HCT116 p53$^{+/+}$ cells were seeded in 96-well opaque plates at a density of 7500 cells/well. 24 h after seeding, cells were treated for 20 h with a range of concentrations of [$L_2Cu_3$]$^{6+}$, [$L_2Zn_3$]$^{6+}$, [$L_2Mn_3$]$^{6+}$, or vehicle control (2-fold serial dilution from 100-3.125 μM) before determining total cellular ATP content by quantification of luminescence (Cell-TiterGlo® assay, Promega). ATP levels were normalised relative to levels in vehicle control-treated cells.

**Cell treatment for preparation of whole cell protein lysates**. ARPE-19 and HCT116 p53$^{+/+}$ cells were seeded at a density of $3 \times 10^5$ (ARPE-19) or $5 \times 10^5$ (HCT116 p53$^{+/+}$) in T25 flasks. 24 h after seeding, cells were treated with 10 μM of [$L_2Cu_3$]$^{6+}$, [$L_2Zn_3$]$^{6+}$, or vehicle control for 4 h (Fig. 8b) or 5 μM of [$L_2Cu_3$]$^{6+}$, [$L_2Zn_3$]$^{6+}$, or vehicle control for 40 h (Fig. 10). Cells were then harvested and lysed in RIPA buffer (Sigma) with the protein concentration of lysates determined by the BCA assay.

**Immunoblotting**. Protein lysates and recombinant proteins were resolved on 15% SDS polyacrylamide gels. Proteins were electroblotted onto nitrocellulose membrane by wet transfer in 1xTris Glycine buffer (Biorad) at 35 mA overnight at room temperature[62]. After blocking of membranes, these were incubated with primary antibody overnight at 4 °C before addition of rabbit or mouse secondary antibody (HRP-conjugated; 1:5000, Dako P0448 or P0260) for 1 h at room temperature and development of blots by enhanced chemiluminescence. Primary antibodies were: anti-Src (total) (Cell Signalling Technology #2123 1:1000), anti-phosphorylated Src (Y527) (Cell Signalling Technology #21055 1:1000), anti-phosphorylated Src (Y416) (Cell Signalling Technology #69435 1:1000), anti-AMPKα (total) (Cell Signalling Technology #2532 1:1000), anti-phosphorylated AMPKα (T172) (Cell Signalling Technology #2535 1:1000), anti-AMPKβ1 (total) (Cell Signalling Technology #4150 1:1000), anti-phosphorylated AMPKβ1 (S108) (abcam ab156890 1:500), anti-phospho-Tyr (pan) (Cell Signalling Technology #8954 1:2000), anti-p53 (Santa Cruz, DO-1 clone, 1:1000,), anti-phosphorylated LDH-A (Y10) (Cell Signalling Technology #8176 1:1000), anti-LDH-A/B (total) (abcam ab134187 1:5000), anti-β-actin (Merck MAB1501, 1:40,000). Quantification of signals was performed by densitometry using Image J software. Uncropped and unprocessed scans are provided in the Source Data file.

**Reporting summary**. Further information on research design is available in the Nature Research Reporting Summary linked to this article.

## Data availability

The X-ray crystallographic data reported in this study have been deposited at the Cambridge Crystallographic Data Centre (CCDC), under deposition number CCDC 2005221 - 005132, 2005133, 2005186, 2005189 and 2005190. This data can be obtained free of charge from The Cambridge Crystallographic Data Centre via www.ccdc.cam.ac.uk/data_request/cif. The data that supports the findings of this study are available in the Source Data file or from the corresponding authors upon reasonable request. Source data are provided with this paper.

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

## Acknowledgements
The authors would like to thank University of Huddersfield for financial support.

## Author contributions
C.R.R., S.J.A. and R.M.P. conceived and designed the experiments and worked on the manuscript. S.J.A., H.W. and H.B.S.G. performed the mechanistic experiments. C.J.C. synthesis of the complexes and crystal structures. R.A.F. performed NMR and UV–Vis studies. C.R.R. collection of crystallography data. R.M.P., S.J.A. and E.P. cytotoxic studies and other cellular studies. J.O.L. elucidation of the mechanism of action. J.B. analysis of the kinome map. J.H. and M.G. obtained the confocal images.

## Competing interests
The authors declare no competing interests.
