## [Peer Review File · Nature Communications]

Reviewers' Comments:

Reviewer #1:

Remarks to the Author:

Phillips, Rice and the coworkers report the anion binding behaviors and biological activities of some self-assembling trimetallic complexes, $[L_2M_3]^{6+}$ (L is tripodaltrispyridylthiazole, $M = Cu^{2+}$, Zn^{2+} or Mn^{2+}), which provides a cleft to encapsulate a range of anions. The experiments of the hydrolysis of phosphate esters, the mechanisms of inhibition of multiple kinases, and the activities towards cancer cells show very interesting results, which will be helpful to study the anion complexes in the biochemistry. This work can be accepted by the journal of Nature Communications after the following points were addressed.

1. The complexes have shown poor solubility in water. In the cancer cell experiments, are the trimetallic complexes dissolved or just dispersed in the system? If only dispersion, how to evaluate the activities?
2. L313-316, the SO_4 and $PhOPO_3$ complexes are significantly more active than ... $[L_2Zn_3]^{6+}$. The later compound show unusually poor activity. Can you give more explanation?
3. If the trimetallic complexes can bind phosphate ion strongly, as given in the paper, is the hydrolysis of phosphate esters or the reactivity of cancer cells catalytic or stoichiometric?

Reviewer #2:

Remarks to the Author:

This paper reports the synthesis of three tri-metallic complexes (L_2M_3 where $M = Cu$, Mn and Zn – one of them previously reported by the same authors) and their structural characterisation with different anions. Depending on the nature of the metal centre, they observed the ability of the system to hydrolyse phosphor-esters (including phosphorylated amino acids). They studied the cytotoxicity of the complexes against a panel of cancer and non-cancer cell lines, and show that the nature of the metal has a clear effect on the activity. They also present some data that the nature of the anions modulates the activity (although this is less convincing as discussed below). To explain the mechanism of action of these complexes, they have carried out a number of studies including the effect the compound have on the activity of a range of kinases. In the last part of the paper, they show studies to study the ability of the complexes to induce autophagy. The idea of using supramolecular assemblies as potential drugs is appealing, especially if their activity could be carefully modulated by the presence of different metals and anions. Therefore, the studies presented are of interest and topical. The studies have been carefully carried out but not all the data is consistent with the explanations/conclusions. The paper could potentially be suitable for publication but there are several issues that need to be addressed (please see below). Experimental details – how was the purity of the complexes confirmed? While the X-ray crystallography shows the structure of the complexes, the purity of the bulk sample has not been ascertained (e.g. by elemental analyses). Considering the differences in cytotoxicity depending on the metal and the counterion, it is very important to ensure that the samples used for the cellular studies are uniform and pure – otherwise, some of the observed differences could be due to different levels of purity.

In relation to the above: in addition to the encapsulated anion (when there is one) what are the counterions in the complexes? Throughout the paper they are represented as the cationic complex (i.e. no counterions indicated) and in the ESI it is indicated that the counterions are ClO_4 (presumably based on the X-ray structures). This is potentially problematic since it could be that the bulk sample contains a mixture of counter-anions and this would affect the cytotoxicity. The ESI-MS suggests that perchlorates are the counterions but shouldn't this be confirmed by elemental analyses as well?

For the cellular studies, how were the compounds' solutions prepared (I could not find this information in the ESI)? I assume that a buffer was used in which case there would be different anions in solution and hence might make the anion in the complex irrelevant in terms of activity (e.g. if phosphate buffer was used, it would likely exchange the anion in the complex with phosphate anyway). Please specify this and comment.

The authors propose that the Zn and Mn cryptands are responsible for the hydrolysis of the $PhOPO_3$. However, there is no direct evidence that this happens with the cryptands rather than

with either mono-metallic ML complexes or even the free cations – in the case of Zn(II), it is well known to act as Lewis acid for the catalytic hydrolysis of phosphoesters. Did the authors carry out the controls using the monometallic Zn, Mn and Cu complexes? See below for comments about this.

Page 8, line 225 – there is an O missing from Serine-PO₃²⁻

For the hydrolysis of phospho-serine (Figure 5): why do you observe a triplet rather than a singlet at ca. 3 ppm (is the ³¹P NMR 1H-coupled or decoupled)?

Figures 6b, 6c and 6e – there are no error bars in these three plots (unlike for 6a and 6d). Please clarify this (caption says that all measurements are average of 3 runs and hence an error bar should be shown).

Page 11 – the authors show the compounds have inhibitory activity against some kinases. They propose that this could be due to either dephosphorylation processes or binding/activity against ATP (although they discard this on the basis of their spectroscopic *in vitro* data). However, they should consider another possibility: the complexes could inhibit the kinases via direct binding (there are previously reported examples of metal complexes binding to the active sites of kinases and inhibiting their activity – see e.g. Megger's work). The authors suggest that there might be some interaction with phosphorylated sites in the kinases but they do not suggest the possibility of binding to the active site leading to inhibition. Can they discard this possibility (especially if the compounds disassemble in solution – see below)?

In relation to the activity of the compounds (in the *in vitro* assays such as kinases or the cytotoxicity): the authors have assumed that the trimetallic assemblies remain intact in solution. However, under the conditions of the assays (and certainly in a cellular environment) there might be many competing ligands that could break the assemblies and the observed activity would not be due to the cryptands but fragments of them. Why didn't they study this possibility? For example, the assays should have been carried out as well with the monometallic compounds. They might display the same activity (normalised to the number of metal centres) than the cryptands. Minor issue – It would be helpful to include in the main text (not the ESI) a scheme showing the chemical structure of the ligand and the products obtained with the different metals/stoichiometries.

Reviewer #3:

Remarks to the Author:

The authors report that Cu and Zn trimeric cryptands are especially potent inhibitors of cancer cell lines. They postulate that this is due to their ability to bind or hydrolyse phosphate esters.

This reviewer's comments will be restricted to the protein phosphorylation aspects of the manuscript.

1 Line 191 Fig 5. Showing the Phosphatase activity of [L₂Zn₃]⁶⁺. There several puzzling aspects to the results. Why do PhOPO₃ and phosphotyrosine behave so differently? PhOPO₃ undergoes a stoichiometric dephosphorylation over 44 hrs whereas phosphotyrosine disappears at t₀. Why do the doublet ³¹P signals only appear in the phosphoserine sample. Since the phosphorylated amino acids are present in proteins and the properties of the phosphorylated residues in proteins differ from the isolated phosphoamino acids eg phosphoserine in proteins is alkali labile whereas isolated phosphoserine is not. To be more relevant this experiment needs to be done with at least phosphopeptides for each phosphor-amino acid. In terms of the biology a large number of kinases phosphorylate Ser-Pro sequences, it would seem imperative to check at least Pro on either side of the phospho-amino acid as was done for alkali-lability some years ago. The experimental conditions are not adequately described to allow someone to repeat the experiments eg the concentration of HEPES is not given. Since HEPES (4-(2-hydroxyethyl)-1-piperazineethanesulfonic acid) is a zwitterion why does the sulfonic acid moiety not compete with the phospho-amino acids? Could competition with HEPES explain the phosphothreonine result? What is the evidence that phosphoserine is hydrolysed versus β-eliminated?

2 Line 241 Fig 6/7. The striking differences in the sensitivity of the different cancer cell lines to the compounds begs the question of whether this can be explained by differential uptake. It would greatly strengthen the manuscript if the authors measured the cellular concentration of the

compounds in a few of the sensitive and insensitive cell lines. Otherwise the results represent a phenomena where the most basic question, are the compounds taken up has not been answered? By using cryptan ion precursor scanning they may even be able to infer the nature of the bound anion in the cell.

3 Line 327 Fig 8a.b. Mechanistic studies. The authors have screened the compounds (10 mM) using the Dundee kinase screening service. Typically, these screens are done for kinase active site inhibitors at limiting ATP concentrations. The results shown in Fig 8 represent preliminary data. Some simple kinetics experiments testing whether the compounds are ATP competitive inhibitors are required. This need only be done on a representative kinase (eg AMPK). If they turn out to be non-competitive inhibitors this might mean they act by blocking the release of ADP, the rate limiting step in the kinase reaction. The activation of Src requires at least an attempt to explain the phenomena. What form of Src is used in the screen, is it full length?

4 Line 374 Fig 9. In vitro phosphatase assay. AMPK was incubated with [L2Zn3]6+ and ATP [50 mM] for 4 hrs with loss of pT172. Is ATP essential and what happens if AMP is used does it inhibit pT172 promoted dephosphorylation (see Line 455)? Since no Mg2+ is included the ATP would be expected to act through the γ -subunit. It would be important to know the [L2Zn3]6+ dose response curve and a time course. Recombinant AMPK is typically phosphorylated on multiple sites, is dephosphorylation restricted to pT172 or are other sites also dephosphorylated (eg β -S108) In Fig 11 with cultured HCT116 cells there is enhanced pT172 signal with [L2Cu3]6+ and [L2Zn3]6+ treatment? The Fig 9 result for Src shows increased phosphorylation of pY419 in the presence of [L2Zn3]6+ but not with [L2Cu3]6+ and yet in Fig S9 the greatest stimulation of Src activity is seen with [L2Cu3]6+.

5 Fig 10b. What is the effect of [L2Zn3]6+ on glycolysis and respiration in ARPE-19 and HCT116 cells. This maybe more significant on ATP levels than the speculated futile cycles of protein phosphorylation and dephosphorylation.

Overall

The authors have provided some very interesting observations on the effects of trimetallic cryptands on protein kinases and cancer cell lines. At present the observations are very much "phenomena" lacking sufficient mechanistic detail. There are too many unanswered questions but these could be tackled by simple experiments as outlined above.

Self-Assembly of an Anion Receptor with Metal-dependent Kinase Inhibition and Potent *In Vitro* Anti-Cancer Properties.

Simon J. Allison^{1*}, Jaroslaw Bryk¹, Christopher J. Clemett¹, Robert A. Faulkner¹, Michael Ginger¹, Hollie B. S. Griffiths¹, Jane Harmer¹, P. Jane Owen-Lynch¹, Emma Pinder¹, Heiko Wurdak², Roger M. Phillips^{1*} and Craig R. Rice^{1*}

Response to Reviewers comments

Reviewer #1 (Remarks to the Author):

Phillips, Rice and the coworkers report the anion binding behaviours and biological activities of some self-assembling trimetallic complexes, $[L_2M_3]^{6+}$ (L is tripodaltrispyridylthiazole, M = Cu^{2+} , Zn^{2+} or Mn^{2+}), which provides a cleft to encapsulate a range of anions. The experiments of the hydrolysis of phosphate esters, the mechanisms of inhibition of multiple kinases, and the activities towards cancer cells show very interesting results, which will be helpful to study the anion complexes in the biochemistry. This work can be accepted by the journal of Nature Communications after the following points were addressed.

R#1 C1. The complexes have shown poor solubility in water. In the cancer cell experiments, are the trimetallic complexes dissolved or just dispersed in the system? If only dispersion, how to evaluate the activities?

Author response: The trimetallic complexes are initially dissolved in DMSO (at 50mM concentration) and complexes are then further diluted in aqueous cell culture media to working solutions and a non-toxic final DMSO concentration of 0.2% (used as vehicle control) without precipitating out of solution. In Section B (subsection ‘Cell seeding for chemosensitivity studies’) of the submitted ESI, the following text describes how the compounds were reconstituted: “[L_2M_3]⁶⁺ and all the other self-assembling complexes were freshly formed by adding DMSO to individual components and mixing together by pipetting. These were then further diluted in cell culture media such that the final DMSO concentration that cells were exposed to was 0.2% (vehicle control).”

R#1 C2. L313-316, the SO_4^{2-} and $PhOPO_3^{2-}$ complexes are significantly more active than [L_2Zn_3]⁶⁺. The later compound shows unusually poor activity. Can you give more explanation?

Author response: The encapsulated anions have two possible consequences: firstly the charge state of the complex is altered and secondly as the cavity is already occupied this will change the rate of encapsulation of other anions (viz the phosphorylated biological species) with the latter probably the most important effect. The reviewer refers to activity against the PSN1 cancer cell line (lines 313-316) and is absolutely right in pointing out that SO_4^{2-} and $PhOPO_3^{2-}$ complexes are significantly more active towards this line than [L_2Zn_3]⁶⁺ which showed a comparatively modest IC_{50} of ~9-10uM. In contrast, however, against several other cancer cell lines, the SO_4^{2-} and $PhOPO_3^{2-}$ complexes are *less* active than [L_2Zn_3]⁶⁺ indicating that effects are complex and cell line dependent. We think that the different genetic and epigenetic backgrounds (oncogenic lesions) of the different cancer cell lines and associated effects of this on cellular dependencies and kinase signalling contribute to the different

responses observed. Further mechanistic understanding of this, however, is beyond the scope of the current manuscript and would likely require multiple ‘-omics’ based approaches (eg. phosphoproteomics with KSEA, WGA and gene expression analyses of individual lines). We do however acknowledge that ‘the mechanistic basis for these differential effects requires further investigation’ on page 12 and 13 of the revised manuscript.

R#1 C3. If the trimetallic complexes can bind phosphate ion strongly, as given in the paper, is the hydrolysis of phosphate esters or the reactivity of cancer cells catalytic or stoichiometric?

Author response: This is a very good point. It would certainly seem that with both $[L_2Cu_3]^{6+}$ and $[L_2Zn_3]^{6+}$ the reaction is stoichiometric (for the latter NMR studies confirm this result ESI – Figure S4.5). However, there is some evidence to suggest that $[L_2Mn_3]^{6+}$ demonstrates some catalytic activity. The mechanism and mode of action is complex due to the two rates of reaction but even once encapsulated the Mn^{2+} complex shows some activity albeit substantially reduced (ESI – Figure S5).

Reviewer #2 (Remarks to the Author):

This paper reports the synthesis of three tri-metallic complexes ($[L_2M_3]^{6+}$ where M = Cu, Mn and Zn – one of them previously reported by the same authors) and their structural characterisation with different anions. Depending on the nature of the metal centre, they observed the ability of the system to hydrolyse phospho-esters (including phosphorylated amino acids). They studied the cytotoxicity of the complexes against a panel of cancer and non-cancer cell lines, and show that the nature of the metal has a clear effect on the activity. They also present some data that the nature of the anions modulates the activity (although this is less convincing as discussed below). To explain the mechanism of action of these complexes, they have carried out a number of studies including the effect the compound has on the activity of a range of kinases. In the last part of the paper, they show studies to study the ability of the complexes to induce autophagy.

R#2 C1. The idea of using supramolecular assemblies as potential drugs is appealing, especially if their activity could be carefully modulated by the presence of different metals and anions. Therefore, the studies presented are of interest and topical. The studies have been carefully carried out but not all the data is consistent with the explanations/conclusions. The paper could potentially be suitable for publication but there are several issues that need to be addressed (please see below). Experimental details – how was the purity of the complexes confirmed? While the X-ray crystallography shows the structure of the complexes, the purity of the bulk sample has not been ascertained (e.g. by elemental analyses). Considering the differences in cytotoxicity depending on the metal and the counterion, it is very important to ensure that the samples used for the cellular studies are uniform and pure – otherwise, some of the observed differences could be due to different levels of purity.

Author response:—Elemental analysis of the lead complexes are now provided in the ESI demonstrating their purity (synthesis of complexes pages 2 and 3 of the ESI).

Additionally, for the chemosensitivity assay experiments, for each independent biological repeat multiple cell lines were analysed in parallel using the same freshly reconstituted complex which should eliminate any possible issues of differences in uniformity or purity of complex that is exposed to different cell lines.

R#2 C2. In relation to the above: in addition to the encapsulated anion (when there is one) what are the counterions in the complexes? Throughout the paper they are represented as the cationic complex (i.e. no counterions indicated) and in the ESI it is indicated that the counterions are ClO₄⁻ (presumably based on the X-ray structures). This is potentially problematic since it could be that the bulk sample contains a mixture of counter-anions and this would affect the cytotoxicity. The ESI-MS suggests that perchlorates are the counterions but shouldn't this be confirmed by elemental analyses as well?

Author response: Thank you for this important point. For the chemical studies either ClO₄⁻, OTf⁻ or BF₄⁻ were used in the crystals growth studies (as different counter anions can give varying quality of crystals; please see ESI section 1 for elemental analyses and section 3 for crystallography) and in the ESI-MS studies ClO₄⁻ was used. For all the biological studies acetate was used as the toxicity of the former anions (as the referee correctly indicates) could play a factor so acetate was used in place of the other three. Previous work shows that the complexes are formed independent of monoanionic counter-anion (e.g. non-encapsulated (or very weakly encapsulated) counter-anions such as perchlorate, tetrafluoroborate, acetate and nitrate see Rice *et. al.* *Angewandte Chemie* 130 (40), 13255-13259). Interestingly, some comparative biological analysis was purposefully carried out on both [L₂M₃](Ac)₆ and [L₂M₃](BF₄)₆ and these showed little difference. Against DLD1 cancer cells, [L₂M₃](Ac)₆ resulted in a mean IC₅₀ of 0.17 μM ±0.06 and [L₂M₃](BF₄)₆ resulted in a mean IC₅₀ of 0.18 μM ±0.07. In contrast, against the ARPE19 non-cancer cells, [L₂M₃](Ac)₆ resulted in a mean IC₅₀ of 157.25 μM ±10.01 and [L₂M₃](BF₄)₆ resulted in a mean IC₅₀ of >100 μM from three independent repeats (concentrations over 100 μM were not tested).

R#2 C3. For the cellular studies, how were the compounds' solutions prepared (I could not find this information in the ESI)?

Author response: see comment R#1 C1 above.

R#2 C4. I assume that a buffer was used in which case there would be different anions in solution and hence might make the anion in the complex irrelevant in terms of activity (e.g. if phosphate buffer was used, it would likely exchange the anion in the complex with phosphate anyway). Please specify this and comment.

This is another very interesting point and we thank the referee for bringing it to our attention. Yes, in the buffered cell culture media in which the complexes are diluted, phosphate is present and in enough quantity to be fully encapsulated within the complex. It is therefore likely that in this media [L₂Cu₃]⁶⁺ will encapsulate phosphate and form [L₂Cu₃(PO₄)]³⁺. With pre-encapsulated complexes (e.g. [L₂Cu₃(SO₄)]⁴⁺ and [L₂Cu₃(O₃POPh)]⁴⁺) displacement with phosphate is theoretically possible and it's likely that a mixture of anion encapsulated species may result (dependent upon pH see Rice *et. al.* *Angewandte Chemie* 132 (46), 20660-20664 and 130 (40), 13255-13259). However, this of particular importance for the zinc-containing species. As we have demonstrated that without the presence of tetrahedral mono-anions, reaction of the ligand with zinc forms the mononuclear species – using a buffer containing phosphate will mean that in the chemosensitivity experiments [L₂Zn₃(PO₄)]³⁺ will be assembled even though just ligand and metal ion are added. Close examination of Figure 7, which demonstrates the effect of the anion on potency and selectivity, strongly supports this assumption. The data shown in that figure is a comparison of “[L₂Zn₃]⁶⁺” with anion encapsulated complexes and it almost all cases it clearly shows that “[L₂Zn₃]⁶⁺” hardly differs at all with the pre-encapsulated [L₂Zn₃(PO₄)]⁶⁺ complex for both potency and selectivity – indicating in the buffered solution [L₂Zn₃(PO₄)]³⁺ is formed irrespective. Interestingly, the sulfate and phenyl phosphate derivatives do show differences in potency and selectivity supporting that different encapsulated anions (e.g. [L₂Cu₃(SO₄)]⁴⁺ and [L₂Cu₃(O₃POPh)]⁴⁺) are present in the media despite phosphate being present.

R#2 C5. The authors propose that the Zn and Mn cryptands are responsible for the hydrolysis of the PhOPO3. However, there is no direct evidence that this happens with the cryptands rather than with either mono-metallic ML complexes or even the free cations – in the case of Zn(II), it is well known to act as Lewis acid for the catalytic hydrolysis of phosphoesters. Did the authors carry out the controls using the monometallic Zn, Mn and Cu complexes?

Author response: The mononuclear complexes i.e. $[LZn]^{2+}$ are only formed when no tetrahedral oxoanions are present – we have shown that in the presence of these anions the trinuclear species is formed. In the ^{31}P NMR experiments when one equivalent of phosphate ester is reacted with three equivalents of Zn^{2+} and two equivalents of L all of the phosphate ester is consumed and only $[L_2Zn_3(PO_4)]^{3+}$ is present (identical chemical shift compared to a sample prepared directly from $[L_2Zn_3]^{6+}$ and PO_4^{3-} see ESI Fig. S4.1). In an similar experiment where one equivalent of phosphate ester is reacted with one equivalent of Zn^{2+} and one equivalent of L (potentially “forcing” the formation of $[LZn]^{2+}$ (see note below)) not all of the phosphate is consumed and still present is a signal corresponding to $[L_2Zn_3(PO_4)]^{3+}$. This indicates that in despite the stoichiometry in the presence of a phosphate ester $[LZn]^{2+}$ rearranges to $[L_2Zn_3(PO_4R)]$ and free ligand. 1H NMR experiments with free Zn^{2+} shows no hydrolysis of the phosphoesters occurs.

Furthermore we do show that $PhOPO^{2-}$ are encapsulated within the trinuclear assembly for Cu^{2+} , Zn^{2+} , Mn^{2+} , (see ESI S2.11, S2.17 and S2.20 respectively) and these are hydrolysed to PO_4^{3-} for Zn^{2+} and Mn^{2+} (see ESI S2.24 and S2.27) but not for Cu^{2+} (see ESI S2.23).

Furthermore, cytotoxic studies using one equivalent of both ligand and metal (forcing the mononuclear complex) shows the same potency but significantly reduced selectivity. It is highly probable that cytotoxicity investigations at this stoichiometry results in the trinuclear complex ($[L_2M_3]^{6+}$) and “free” ligand (see note below). Chemosensitivity data on the free ligand shows that whilst it is highly toxic to cancer cell lines it shows vastly reduced selectivity (see ESI Fig. S9) and it is seems probable that “free” ligand removes metal ions from the biological systems and is consequently toxic.

It is worth pointing out that the Cu^{2+} complex is trinuclear even without the presence of oxoanions and its biological activity is broadly similar to that of the zinc species.

Note. Forcing the formation of $[LM]^{2+}$ by controlling the stoichiometry *could* result in redistribution to the trinuclear complex (e.g. $3[ML]^{2+} - [L_2M_3]^{6+}$ plus 1L and this definitely occurs in the presence of oxoanions (e.g. $3[ML]^{2+}$ plus EO_4^{n-} goes to $[L_2M_3(EO_4)]^{(6-n)+}$ plus 1L). With phosphate present in the buffer this seems likely.

R#2 C6. Page 8, line 225 – there is an O missing from Serine-PO32-

Thank you – now corrected

R#2 C7. For the hydrolysis of phospho-serine (Figure 5): Why do you observe a triplet rather than a singlet at ca. 3 ppm (is the ^{31}P NMR 1H-coupled or decoupled)?

The ^{31}P NMR of the phospho-esters are 1H -coupled whereas the experiments are decoupled. Figure amended accordingly.

R#2 C8. Figures 6b, 6c and 6e – there are no error bars in these three plots (unlike for 6a and 6d). Please clarify this (caption says that all measurements are average of 3 runs and hence an error bar should be shown).

The selectivity indices (SI) are generated using the mean IC50 values for which error bars are provided. As such the experimental error is presented in the IC50 data set as there is only one value for the SI. The lack of error bars for SI data is explained in the legend to figure 6 and we hope this will provide the reader with an explanation for the lack of error bars in figure 6b, c and e.

R#2 C9. Page 11 – the authors show the compounds have inhibitory activity against some kinases. They propose that this could be due to either dephosphorylation processes or binding/activity against ATP (although they discard this on the basis of their spectroscopic in vitro data). However, they should consider another possibility: the complexes could inhibit the kinases via direct binding (there are previously reported examples of metal complexes binding to the active sites of kinases and inhibiting their activity – see e.g. Megger's work). The authors suggest that there might be some interaction with phosphorylated sites in the kinases but they do not suggest the possibility of binding to the active site leading to inhibition. Can they discard this possibility (especially if the compounds disassemble in solution – see below)?

We thank the reviewer for these useful comments. We now directly show that neither $[L_2Cu_3]^{6+}$ nor $[L_2Zn_3]^{6+}$ is either, i) an ATP competitive inhibitor or, ii) a substrate competitive inhibitor of recombinant human AMPK. These results suggest that the complexes do not directly bind to the active site of AMPK and are presented in the new Fig. 9 and discussed on page 13 of manuscript. The text also has been further modified on page 15 to acknowledge that multiple mechanisms may contribute to the observed kinase inhibition (or indeed activation for several kinases) as indicated below:

“..or through steric or structural effects. One possible mechanism of kinase inhibition by either complex not formally investigated is the blocking of ADP release or it may be that multiple mechanisms contribute to the observed kinase inhibition or activation depending on the particular kinase.”

R#2 C9. In relation to the activity of the compounds (in the in vitro assays such as kinases or the cytotoxicity): the authors have assumed that the trimetallic assemblies remain intact in solution. However, under the conditions of the assays (and certainly in a cellular environment) there might be many competing ligands that could break the assemblies and the observed activity would not be due to the cryptands but fragments of them. Why didn't they study this possibility? For example, the assays should have been carried out as well with the monometallic compounds. They might display the same activity (normalised to the number of metal centres) than the cryptands.

Please see our detailed response to comment 5 (R#2 C5) which addresses these comments. As detailed there, we have now performed the chemosensitivity assays with monometallic compounds as suggested (please see ESI Fig. S9) and with ligand alone, however, selectivity towards cancer cells is substantially decreased. We postulate that might be due to removal of Zn^{2+} (or other d-block metal ions) from the cellular environment due to rearrangement of the stoichiometrically forced mononuclear complex to trinuclear complex ($[L_2M_3]^{6+}$) and “free” ligand.

R#2 C10. Minor issue – It would be helpful to include in the main text (not the ESI) a scheme showing the chemical structure of the ligand and the products obtained with the different metals/stoichiometries.

Chemical structure and scheme added.

Reviewer #3 (Remarks to the Author):

The authors report that Cu and Zn trimeric cryptans are especially potent inhibitors of cancer cell lines. They postulate that this is due to their ability to bind or hydrolyse phosphate esters. This reviewer's comments will be restricted to the protein phosphorylation aspects of the manuscript.

R#3 C1. Line 191 Fig 5. Showing the Phosphatase activity of $[L_2Zn_3]^{6+}$. There several puzzling aspects to the results. Why do $PhOPO_3$ and phosphotyrosine behave so differently? $PhOPO_3$ undergoes a stoichiometric dephosphorylation over 44 hrs whereas phosphotyrosine disappears at t_0 .

Author response: Perhaps we misunderstand the reviewer's comment but the ^{31}P signal for phosphotyrosine doesn't disappear at $t = 0$, and a trace is still observable (-1.5 ppm) after 44 hrs in a remarkably similar fashion to $PhOPO_3^{2-}$ (see Fig 5 C vs O (19 hrs) and D vs P (44 hrs)). The signal does slightly broaden, and this is due to binding of the cryptate (it seems almost certain that initially the phosphate ester will be bound (probably in a similar fashion to $[L_2Cu_3(O_3POPh)]^{4+}$ and this binding will give a broadened spectra (due to restricted rotation, desymmetrising of the cryptate and formation of diastereoisomers with the racemic cage and stereopure "bound" phosphate). This is not observed to the same degree in $PhOPO_3^{2-}$ as it is achiral and does not contain sp^3 substituents. The differential broadness is why direct comparative integration has not been carried out. This is further supported by the signal at 8.6 ppm (corresponding to $[L_2Zn_3(PO_4)]^{3+}$) which clearly grows over time (see Fig 5 N, O and P).

Exploded view of Figure 5 showing the hydrolysis of phenyl phosphate (LHS) and phosphotyrosine (RHS).

R#3 C2. Why do the doublet ^{31}P signals only appear in the phosphoserine sample.

Author response: The two signals corresponding to the diastereoisomers are visible for both the phosphoserine and phosphotyrosine encapsulated complexes in the ^{31}P NMR spectra. In all cases the

chemical shifts are different – which would be expected as they are different diastereoisomers. The phosphothreonine is a little more difficult to observe (partly due to the reaction not going to completion and therefore lower concentration of the hydrolysis product) but close examination (especially after 19 hrs Fig 5 K) a shoulder can be seen on the product signal.

Exploded view of Figure 5 showing the two signals corresponding to a mixture of diastereoisomers between the racemic cryptand and the resolved chiral amino acids which will form an ion-pair (e.g. $[\text{L}_2\text{Zn}_3(\text{PO}_4)](\text{MeCH}(\text{NH}_2)\text{CO}_2)^{2+}$); left serine; right tyrosine.

R#3 C3. Since the phosphorylated amino acids are present in proteins and the properties of the phosphorylated residues in proteins differ from the isolated phosphoamino acids eg phosphoserine in proteins is alkali labile whereas isolated phosphoserine is not. To be more relevant this experiment needs to be done with at least phosphopeptides for each phosphor-amino acid. In terms of the biology a large number of kinases phosphorylate Ser-Pro sequences, it would seem imperative to check at least Pro on either side of the phospho-amino acid as was done for alkali-lability some years ago.

Author response:

Evidence is provided that $[\text{L}_2\text{Zn}_3]^{6+}$ can partially dephosphorylate phosphorylated Thr172 of recombinant human AMPK α 1 protein (Fig. 10, Suppl. Fig.11), phosphorylated Tyr527 of recombinant human Src (Fig. 10) and also Ser108P of AMPK β 1 (new Suppl. Fig.11) consistent with the inhibition of AMPK kinase in the kinase screen and the activation of Src (Fig.8). New data also indicates decreased phospho-Thr172 AMPK α in HCT116 cancer cells (Fig. 11).

The amino acid sequences immediately surrounding AMPK α T172, Src Y527, and AMPK β 1 S108 (phosphorylated amino acid indicated in bold and by an asterisk) are as follows:-

SDGEFLRT*SCGSPNY (T172 AMPK α)

TSTEPQY*QPGENL (Y527 Src)

SKLPLTRS*HNNFVAI (S108 AMPK β 1)

None of these amino acid sequences have a Pro directly adjacent to the phosphorylated amino acid or appear to share a common phosphorylation motif. However, over 1400 phosphorylation motifs have been identified to date (Sugiyama, N., Imamura, H. & Ishihama, Y. Large-scale Discovery of Substrates of the Human Kinome. Sci Rep 9, 10503 (2019)). As such elucidation of the phosphorylation motif preferences of $[\text{L}_2\text{Zn}_3]^{6+}$ or towards phosphorylated Ser-Pro sequences is beyond the scope of the current manuscript.

R#3 C4. The experimental conditions are not adequately described to allow someone to repeat the experiments e.g. the concentration of HEPES is not given.

Author response: Thank you for spotting that omission. HEPES is 60 mmol – all figures and experimental corrected.

R#3 C5. Since HEPES (4-(2-hydroxyethyl)-1-piperazineethanesulfonic acid) is a zwitterion why does the sulfonic acid moiety not compete with the phospho-amino acids?

Author response: The ability of the cryptate to encapsulate anions has a strong correlation to the anionic charge. As the HEPES is a monoanionic sulfonate ester it may bind the cavity, but this would be weakly compared to tri- or di-anionic anions. A ^{31}P NMR of a solution of $[\text{L}_2\text{Zn}_3]^{6+}$ plus $(\text{Bu}_4\text{N})\text{H}_2\text{PO}_4$ in HEPES (approximately 10-fold excess of HEPES to complex) shows only a signal at ~ 9 ppm corresponding to $[\text{L}_2\text{Zn}_3(\text{PO}_4)]^{3+}$ and no free phosphate (~ 0 ppm) which would result from competition with HEPES (see ESI Fig 4.4).

R#3 C6. Could competition with HEPES explain the phosphothreonine result? What is the evidence that phosphoserine is hydrolysed versus β -eliminated?

Author response: HEPES is used in all the ^{31}P NMR experiments so the competitive nature of the buffer would apply to all the phosphorylated amino acids. In all cases the phosphate is removed as an encapsulated anion e.g. $[\text{L}_3\text{Zn}_2(\text{PO}_4)]^{3+}$ (due to the identical chemical shift at -10ppm) and we assume that they hydrolyse to the alcohols – it is possible that phosphothreonine undergoes β -elimination but the rate of reaction is slower than the other three examples.

R#3 C7. Line 241 Fig 6/7. The striking differences in the sensitivity of the different cancer cell lines to the compounds begs the question of whether this can be explained by differential uptake. It would greatly strengthen the manuscript if the authors measured the cellular concentration of the compounds in a few of the sensitive and insensitive cell lines. Otherwise the results represent a phenomena where the most basic question, are the compounds taken up has not been answered? By using cryptan ion precursor scanning they may even be able to infer the nature of the bound anion in the cell.

We thank the reviewer for this suggestion. Given the complex chemical composition of the cell following careful consideration we felt that detection of any cryptand-derived m/z ion against a background of cellular material would be extremely difficult and unlikely to provide an answer as to whether the compounds are taken up by cells or whether they may act at the cell membrane/outside of the cell. As an alternative approach to addressing this question, we have utilised the Zn^{2+} reporter dye TSQ (Carpenter, M.C., Lo, M.N., Palmer, A. E. Techniques for measuring cellular zinc. *Archives of Biochemistry and Biophysics*, **611**, 20–29 (2016)) as a surrogate marker for cellular uptake of $[\text{L}_2\text{Zn}_3]^{6+}$ in the HCT116 cancer cells and the ARPE19 non-cancer cells. The results are presented in Figure S16 and they show (i) that $[\text{L}_2\text{Zn}_3]^{6+}$ which is our lead compound is taken up by cells and (ii) there is little difference in the increased TSQ signal (following 1h treatment of cells with $[\text{L}_2\text{Zn}_3]^{6+}$) between the chemoresistant ARPE19 cell line and the sensitive HCT116 cell line. Whilst further studies are required to fully validate these findings, the results suggest that differential response is not due to differences in drug uptake but complex/target interactions and subsequent downstream events.

R#3 C8. Line 327 Fig 8a.b. Mechanistic studies. The authors have screened the compounds (10 μM) using the Dundee kinase screening service. Typically, these screens are done for kinase active site inhibitors at limiting ATP concentrations. The results shown in Fig 8 represent preliminary data. Some simple kinetics experiments testing whether the compounds are ATP competitive inhibitors are required. This need only be done on a representative kinase (eg AMPK). If they turn out to be non-competitive inhibitors this might mean they act by blocking the release of ADP, the rate limiting step in the kinase reaction.

We thank the reviewer for these useful comments which strengthen the manuscript. These experiments have now been performed using AMPK as a representative kinase as suggested. ATP competition assays were performed using 5 different ATP concentrations (0.5, 2, 20 (Km), 200 and 500 μM ATP). The results show that neither $[\text{L}_2\text{Cu}_3]^{6+}$ nor $[\text{L}_2\text{Zn}_3]^{6+}$ (each tested at 10 μM) is an ATP

competitive inhibitor (see new Fig. 9). We further show that neither is a substrate competitive inhibitor of recombinant human AMPK. These results suggest that the complexes do not directly bind to the active site of AMPK and this is discussed on page 13 of manuscript. The text also has been further modified on page 15 to acknowledge that multiple mechanisms may contribute to the observed kinase inhibition (or indeed activation for several kinases). We specifically also now mention the possibility that release of ADP may be blocked (page 15):

“⁴² or through steric or structural effects. One possible mechanism of kinase inhibition by either complex not formally investigated is the blocking of ADP release or it may be that multiple mechanisms contribute to the observed kinase inhibition or activation depending on the particular kinase.”

R#3 C9. The activation of Src requires at least an attempt to explain the phenomena. What form of Src is used in the screen, is it full length?

Please see page 15 & 16 (last paragraph of page 16 – copied below) and accompanying experimental data for Src in Fig.10 which provides a likely mechanism for the observed activation of Src.

“For Src, effects of the complexes on the key regulatory phospho- amino acids Y527 and Y416 were examined (Fig. 10). Phosphorylation at Y527 in the Src C-terminal domain is known to decrease Src activity whereas autophosphorylation at Y416 in the activation loop of the kinase domain increases Src activity.^{40,45} Here, incubation of recombinant Src with $[\text{L}_2\text{Zn}_3]^{6+}$ resulted in a ~30% decrease in p-Y527 relative to total Src levels and a ~5-fold increase in p-Y416 levels (Fig. 10), both of which combined would be expected to result in enhanced Src kinase activity as was observed in the kinase screen (Fig. 8c,d). The decrease in p-Y527 is consistent with selective phosphatase activity of $[\text{L}_2\text{Zn}_3]^{6+}$. Y527 dephosphorylation reduces allosteric inhibition of Src kinase activity by the C-terminal domain, enabling autophosphorylation of Src at Y416.⁴⁰ Similar to AMPK, incubation of $[\text{L}_2\text{Cu}_3]^{6+}$ with Src resulted in higher molecular weight bands detected with the total Src antibody which were not observed with control or $[\text{L}_2\text{Zn}_3]^{6+}$ treatments, suggesting binding of $[\text{L}_2\text{Cu}_3]^{6+}$ to Src. p-Y416 could not be detected raising the possibility that $[\text{L}_2\text{Cu}_3]^{6+}$ binding to Src masks detection of this epitope, however, further studies are required to unravel the detailed mechanism as to how $[\text{L}_2\text{Cu}_3]^{6+}$ binding to Src may increase its activity.”

The recombinant Src used in the kinase screen and the experiment in Fig.10 is full length human Src (536 amino acids) with a N-terminal His tag enabling its purification by nickel agarose following its recombinant expression using baculovirus expression system.

R#3 C10. Line 374 Fig 9. In vitro phosphatase assay. AMPK was incubated with $[\text{L}_2\text{Zn}_3]^{6+}$ and ATP [50 mM] for 4 hrs with loss of pT172. Is ATP essential and what happens if AMP is used does it inhibit pT172 promoted dephosphorylation (see Line 455)?

This has now been tested, both for dephosphorylation of AMPK α at Thr172 and for dephosphorylation of AMPK β 1 at S108. ATP (or AMP) appears to be required - both for dephosphorylation of AMPK α at Thr172 and for dephosphorylation of AMPK β 1 at S108. This data is presented in Supplementary Fig.S12 and is discussed in under ‘phosphatase assay results’ (page 31 of the ESI).

“ Effects of $[\text{L}_2\text{Zn}_3]^{6+}$ on p-T172 and p-S108 appeared to be dependent on the presence of ATP or AMP (ESI Fig. S12). The ability of $[\text{L}_2\text{Zn}_3]^{6+}$ to modestly decrease p-T172 levels in the presence of

AMP is particularly important as AMP binding to the regulatory γ AMPK subunit is known to help protect p-T172 from physiological phosphatases. Furthermore, altering AMP concentration (0-160 μ M) had no effect on the percentage inhibition of AMPK kinase activity by $[\text{L}_2\text{Zn}_3]^{6+}$ (ESI Fig. S12).”

Additionally, we show that the presence of increasing concentrations of AMP (0-160 μ M) has no effect on level of inhibition of AMPK kinase by $[\text{L}_2\text{Zn}_3]^{6+}$ (Fig. S12c).

R#3 C11. Since no Mg^{2+} is included the ATP would be expected to act through the γ -subunit. It would be important to know the $[\text{L}_2\text{Zn}_3]^{6+}$ dose response curve and a time course.

We have performed these experiments as requested and this presented in Fig. S11. Decreases in phosphorylated levels of T172 of AMPK α and phospho-S108 of AMPK β 1 is evident from 2 hours at both phospho-sites. Similarly, a $[\text{L}_2\text{Zn}_3]^{6+}$ concentration titration experiment is presented with dephosphorylation evident from concentrations of 12.5 μ M or higher. The relatively slow kinetics and concentrations needed suggest that there be additional mechanisms of inhibition of AMPK kinase (e.g. via other phospho-sites or steric effects e.g. ADP release as the reviewer mentions). However, importantly we now also show that 4h 10 μ M $[\text{L}_2\text{Zn}_3]^{6+}$ treatment causes a decrease in p-T172 AMPK levels selectively in HCT116 cancer cells (new Fig. 11). This could result from the dephosphorylation of AMPK by $[\text{L}_2\text{Zn}_3]^{6+}$ and/or inhibition of upstream kinases.

R#3 C12. Recombinant AMPK is typically phosphorylated on multiple sites, is dephosphorylation restricted to pT172 or are other sites also dephosphorylated (eg β -S108)

We thank the reviewer for this suggestion and have now tested this for phospho-S108 AMPK β 1 for which antibodies were available to purchase to be able to do this experiment. As shown in Supp.Fig.11& 12 and discussed in response to earlier comments, $[\text{L}_2\text{Zn}_3]^{6+}$ results in a modest decrease in phospho-S108 in a concentration- and time-dependent manner. It is reported in the literature that phosphorylation of S108 of AMPK β 1 promotes AMPK activity so these effects are, together with decreased p-T172, consistent with the observed inhibition of AMPK kinase activity by $[\text{L}_2\text{Zn}_3]^{6+}$ (Fig.8 and new Fig.9)

The text has been modified accordingly:-

“These effects of $[\text{L}_2\text{Zn}_3]^{6+}$ are not restricted to p-T172 of AMPK α , with phosphorylated levels of S108 of the non-catalytic AMPK β subunit also decreasing following incubation with $[\text{L}_2\text{Zn}_3]^{6+}$ in a time- and concentration- dependent manner (ESI Fig. S11). Phosphorylation of S108 of AMPK β increases AMPK activity indicating at least two potential mechanisms by which $[\text{L}_2\text{Zn}_3]^{6+}$ can inhibit AMPK kinase activity (Fig. 8a, 9).”

R#3 C13. In Fig 11 with cultured HCT116 cells there is enhanced pT172 signal with $[\text{L}_2\text{Cu}_3]^{6+}$ and $[\text{L}_2\text{Zn}_3]^{6+}$ treatment?

This is correct for $[\text{L}_2\text{Zn}_3]^{6+}$ but pT172 signal with $[\text{L}_2\text{Cu}_3]^{6+}$ is similar to control treated HCT116 cells (Fig. 11 in the original submission is now Fig.14 in the revised ms). Whilst this may initially seem confusing given that we present data that $[\text{L}_2\text{Zn}_3]^{6+}$ and $[\text{L}_2\text{Cu}_3]^{6+}$ inhibit AMPK kinase (Fig. 8, 9) and $[\text{L}_2\text{Zn}_3]^{6+}$ causes a decrease in pT172 of recombinant AMPK (Fig.10), it is consistent with the established physiological role of AMPK as a sensor of cellular ATP depletion which we also demonstrate and the timings/kinetics of this ATP depletion (Fig. 12). The cellular p-T172 levels are enhanced with $[\text{L}_2\text{Zn}_3]^{6+}$ treatment at this time point (40h) consistent with ATP depletion stimulating phosphorylation of AMPK at T172 and the observed induction of p53 (Fig. 14) indicating cellular stress. In contrast, at an earlier time point of 4h, $[\text{L}_2\text{Zn}_3]^{6+}$ results in decreased pT172 levels (consistent with its dephosphorylation) which is shown in the new Fig. 11.

These results are explained in the manuscript text (page 20, with reference to Fig.14) as follows:-

“Thus, cellular levels of p-T172 of AMPK α were increased relative to total AMPK α levels specifically in the HCT116 cancer cells by [L₂Zn₃]⁶⁺ but not by [L₂Cu₃]⁶⁺ (Fig. 14). T172 AMPK α phosphorylation is induced by low cellular ATP levels and increased AMP or ADP levels, enabling competitive binding of AMP/ADP to the γ regulatory subunit of AMPK enabling AMPK α phosphorylation by one of the AMPK upstream kinases.⁴³ Thus, paradoxically, [L₂Zn₃]⁶⁺ causes dephosphorylation of T172 AMPK α and kinase inhibition in a cell-free system (Fig. 8a, 9) and in the HCT116 cancer cells with brief exposure (Fig. 11) consistent with its phosphatase activity (Fig. 5) but longer treatment results in increased cellular levels of p-T172 AMPK (Fig. 14) which is attributed to selective ATP depletion (Fig. 12).”

R#3 C14. The Fig 9 result for Src shows increased phosphorylation of pY419 in the presence of [L₂Zn₃]⁶⁺ but not with [L₂Cu₃]⁶⁺ and yet in Fig S9 the greatest stimulation of Src activity is seen with [L₂Cu₃]⁶⁺.

The reviewer is correct in pointing out that whilst both [L₂Zn₃]⁶⁺ and [L₂Cu₃]⁶⁺ stimulate Src activity the stimulation was greater with [L₂Cu₃]⁶⁺. However, this is not in disagreement with the effects observed on pY419 but simply indicates that [L₂Zn₃]⁶⁺ and [L₂Cu₃]⁶⁺ are having their stimulatory effects via different mechanisms. The increase in pY419 with [L₂Zn₃]⁶⁺ is consistent with stimulation arising from observed modest dephosphorylation by [L₂Zn₃]⁶⁺ at pY527 which is reported in the literature to be inhibitory thus enabling Src to autophosphorylate itself at Y419. It is possible that dephosphorylation occurs at other sites of Src also promoting its autophosphorylation at Y419.

Whilst [L₂Zn₃]⁶⁺ is shown to hydrolyse phospho-amino acids and phenylphosphate (e.g. Fig. 5) and dephosphorylate p-T172 and p-S108 of AMPK (e.g. Fig. 10, Fig. S11), the chemical and biological data in the manuscript suggests that [L₂Cu₃]⁶⁺ encapsulates or binds rather than dephosphorylating (e.g. Fig. 3 and ESI, encapsulation of phenylphosphate; Fig. 10 – detection of high molecular weight bands >250kDa with total AMPK antibody and total Src antibody specifically when the recombinant kinases are incubated with [L₂Cu₃]⁶⁺). Overall these results suggest a binding mechanism of action for [L₂Cu₃]⁶⁺ and we hypothesise that [L₂Cu₃]⁶⁺ may bind pY419 of Src or this region of Src allosterically promoting activation of Src but preventing pY419 detection with the pY419 antibody.

This is discussed in the manuscript text as follows:-

“Similar to AMPK, incubation of [L₂Cu₃]⁶⁺ with Src resulted in higher molecular weight bands detected with the total Src antibody which were not observed with control or [L₂Zn₃]⁶⁺ treatments, suggesting binding of [L₂Cu₃]⁶⁺ to Src. p-Y416 could not be detected raising the possibility that [L₂Cu₃]⁶⁺ binding to Src masks detection of this epitope, however, further studies are required to unravel the detailed mechanism as to how [L₂Cu₃]⁶⁺ binding to Src may increase its activity.”

R#3 C15. Fig 10b. What is the effect of [L₂Zn₃]⁶⁺ on glycolysis and respiration in ARPE-19 and HCT116 cells. This maybe more significant on ATP levels than the speculated futile cycles of protein phosphorylation and dephosphorylation.

We thank the review for this important comment. We have now investigated the effects of [L₂Zn₃]⁶⁺ on glycolysis and respiration in these 2 cell lines with the data obtained presented in new Fig. 13. As part B of the figure shows, treatment of the HCT116 cells with 25 μ M [L₂Zn₃]⁶⁺ for 20h substantially decreases both mitochondrial respiration rate and glycolysis. This would be expected to significantly reduce the rate of ATP generation and these metabolic effects of [L₂Zn₃]⁶⁺ are therefore likely to be major contributors to the observed decrease in ATP levels in response to [L₂Zn₃]⁶⁺ treatment that are presented in Fig. 12b. However, it is currently unclear as to whether these effects of [L₂Zn₃]⁶⁺ on glycolysis and respiration are linked to its phosphatase activity and inhibition of multiple kinases (and

potentially other proteins regulated by phosphorylation) or are independent of this. Interestingly, 1h 25 μ M [L₂Zn₃]⁶⁺ treatment resulted in a decrease in glycolytic rate in the HCT116 cancer cells (but was without effect on respiratory rate) and this effect was selective to the HCT116 cancer cells (Fig. 13a).

This data is now discussed in the manuscript as follows:-

“Another possibility, however, for the observed ATP depletion (Fig. 12) is decreased ATP generation through the inhibition of glycolysis and/or mitochondrial respiration by the complexes. 1h treatment with 25 μ M [L₂Zn₃]⁶⁺ caused a modest selective decrease in glycolysis in the HCT116 cancer cells whilst the more metabolically quiescent ARPE19 cells were unaffected (Fig. 13a). 20h treatment substantially impaired both mitochondrial respiration and glycolysis in the HCT116 cancer cells (Fig. 13b), likely contributing to the decrease in ATP levels. It is currently unclear whether the effects of [L₂Zn₃]⁶⁺ on glycolysis and mitochondrial respiration are linked to its phosphatase/kinase inhibitory activity or are via an independent mechanism. A number of key metabolic enzymes are kinases themselves (e.g. hexokinase, pyruvate kinase, phosphoglycerate kinase)⁴⁹ whilst others are regulated by phosphorylation (e.g. pyruvate dehydrogenase, lactate dehydrogenase A (LDH-A)). [L₂Zn₃]⁶⁺ was found to modestly decrease p-Y10 LDH-A in ARPE19 cells and to a lesser extent in HCT116 cells (see ESI, Fig. S18).”

R#3 C16. Overall: The authors have provided some very interesting observations on the effects of trimetallic cryptands on protein kinases and cancer cell lines. At present the observations are very much “phenomena” lacking sufficient mechanistic detail. There are too many unanswered questions but these could be tackled by simple experiments as outlined above.

We thank the reviewer for their helpful and detailed comments and suggestions of further experiments that we could do to add more mechanistic detail to the acknowledged very interesting ‘phenomena’ reported. We now have performed these suggested experiments as outlined in the reviewers’ specific comments above and we think that the new data generated significantly strengthens the manuscript.

Mechanistically, new data is now presented that shows:-

- Neither [L₂Zn₃]⁶⁺ nor [L₂Cu₃]⁶⁺ are ATP competitive kinase inhibitors
 - Neither [L₂Zn₃]⁶⁺ nor [L₂Cu₃]⁶⁺ are substrate competitive kinase inhibitors
- (these results differ from most anti-cancer kinase inhibitors that are in clinical use which are ATP-competitive inhibitors (developed through rational drug design or phenotypically selected for in an ATP based screening assay); instead our results suggest that these highly cancer-selective self-assembling anion cryptands act through a non-competitive allosteric mechanism of kinase inhibition or activation).
- [L₂Zn₃]⁶⁺ is able to dephosphorylate phosphorylated S108 of AMPK β 1 in a time- and concentration manner. This identifies a second regulatory phospho-amino acid of AMPK kinase that is dephosphorylated by [L₂Zn₃]⁶⁺ (in addition to dephosphorylation of pT172 of AMPK α for which new data confirms this original finding. As phosphorylation of both of these amino acids of AMPK is associated with AMPK activation, their dephosphorylation by [L₂Zn₃]⁶⁺ provides at least two potential mechanisms by which [L₂Zn₃]⁶⁺ can inhibit AMPK activity).
 - [L₂Zn₃]⁶⁺ decreases endogenous cellular phospho-T172 AMPK levels of HCT116 cancer cells demonstrating that the inhibition of AMPK activity and dephosphorylation observed in cell-free systems is physiologically relevant and can occur in a cellular context. This cellular decrease is time-dependent (observed at 4h), however, with an increase in pT172 detected at 40h. Whilst the increase at 40h may initially seem contradictory to dephosphorylation of AMPK by [L₂Zn₃]⁶⁺, mechanistically it ‘fits’ with the observed depletion of ATP at 20h (Fig. 12b) and well-established activation of AMPK and its phosphorylation at T172 stimulated by decreased ATP/increased AMP.

- Inhibition of glycolysis and mitochondrial respiration by $[\text{L}_2\text{Zn}_3]^{6+}$ which is likely to significantly contribute to the ATP depletion observed. Further investigation that is beyond the scope of this manuscript is required to decipher whether this is linked to the phosphatase activity of $[\text{L}_2\text{Zn}_3]^{6+}$ (e.g. towards metabolic enzymes) or represents a novel mechanism of action.
- Inhibition of AMPK kinase activity by $[\text{L}_2\text{Zn}_3]^{6+}$ is unaffected by increasing concentrations of AMP which is important as AMP activates AMPK and inhibits the activity of physiological phosphatases towards pT172.
- $[\text{L}_2\text{Zn}_3]^{6+}$ is rapidly taken up by cells (within 1h exposure) but the differential chemosensitivity response of cells is unlikely to be due to differential drug uptake as the TSQ signal after $[\text{L}_2\text{Zn}_3]^{6+}$ exposure is similar in both the chemoresistant ARPE19 non-cancer cells and the chemosensitive HCT116 p53^{+/+} cancer cells.

We again thank all the referees for taking the time to review the manuscript, especially considering the volume of science contained within it. We hope we have addressed all their particular concerns and look forward to their comments.

Regards

Craig R. Rice, Simon Allison and Roger Phillips.

Reviewers' Comments:

Reviewer #1:

Remarks to the Author:

In this work, the authors describe the synthesis and appealing anticancer activity of self-assembling trimetallic complexes, investigate their mechanism of action and identify the putative biomolecular targets, although it is not easy to figure out the mechanism and its biological targets. This work offers good enlightenment to the biological application of anion coordination chemistry. I would recommend acceptance of this manuscript after the following points are addressed.

1. What's the binding ability of the $[L_2M_3]^{6+}$ to PO_4^{3-} or SO_4^{2-} ? What's the real molecular species for $[L_2Zn_3(SO_4)]^{4+}$ or $[L_2Zn_3(O_3POPh)]^{4+}$ in the cell culture environments with high phosphate concentration (mM)?
2. Some explanations should be provided. Why is it different against MCF10A cells? It is noteworthy that $[L_2Zn_3]^{6+}$ pre-encapsulated with PO_4^{3-} resulted in similar potencies, except against MCF10A cells, to $[L_2Zn_3]^{6+}$ alone. Due to significant quantities of phosphate in the cell culture media ($\sim 1\text{mM}$), it is likely that $[L_2Zn_3]^{6+}$ would encapsulate PO_4^{3-} to form $[L_2Zn_3(PO_4)]^{3+}$.
3. Can the complexes be internalized into the cells? Some kinases locate at the cell membrane, thus the complexes may not need to enter the cells to exert the anticancer activity.
4. In the sentence, "Following $[L_2Zn_3]^{6+}$ treatment of HCT116 cancer cells, a small decrease ($\sim 10\%$) in total levels of Tyr- phosphorylated proteins was detected by immunoblotting using a pan phosphotyrosine antibody although there was also evidence of increased p-Y of some proteins (*) (Fig. 14)", how about the reliability of decrease of 10%?

Reviewer #2:

Remarks to the Author:

The authors have addressed properly all the queries raised by this reviewer. Therefore, I recommend publication of this interesting manuscript.

Reviewer #3:

Remarks to the Author:

The authors have fully addressed the points raised by referee 3.

Specific points

C1 $PhOPO_3$ and phosphotyrosine dephosphorylation clarified.

C2 Clarified phosphothreonine diastereoisomers.

C3 The authors provide evidence that $[L_2Zn_3]^{6+}$ can partially dephosphorylate phosphorylated proteins thereby establishing that it works on phosphorylated amino acids within proteins. The authors claim that demonstrating that that $[L_2Zn_3]^{6+}$ acted on pSer-Pro and pThr-Pro sequences was beyond the scope of the study. Given the widespread occurrence of Ser-Pro phosphorylation site targets in signaling pathways the referee considers this is still an important point and should be at least highlighted in the Discussion.

C4 Experimental conditions clarified.

C5 Satisfactory explanation that HEPES does not compete.

C6 The authors need to mention "we assume the phosphorylated amino acids hydrolyse to the alcohols" in the text.

C7 Differential uptake. The authors have adequately investigated this point with the Zn^{2+} reporter dyeTSQ.

C8 The authors have adequately addressed the query that [L2Cu3]6+ nor [L2Zn3]6+ can act as ATP competitive inhibitors.

C9 The authors have adequately explained the Src differential dephosphorylation.

C10 In vitro phosphatase assay. AMPK was incubated with [L2Zn3]6+ ±AMP, ATP. In Fig S12a the control without ATP or AMP addition of [L2Zn3]6+ gives an apparent increase in pT172. Across the control and + AMP the addition of [L2Zn3]6+ reduces the loading control for AMPK ie possibly precipitating the AMPK. In S12b the results do not appear to show any effect of ATP and AMP on pS108 dephosphorylation. In S12c at 80 microM AMP there is less AMPK but more at 160 microM. Is AMP at 160 microM protecting the enzyme from precipitation as queried in S12a. Overall, the results do not appear to be informative.

C11 [L2Zn3]6+ dose response curve and a time course. Again, in S11a there appears to be a loss of the total AMPK with [L2Zn3]6+ incubation. But in S11b the total beta1 incubated with [L2Zn3]6+ looks fine. However, the control pS108 values over the 4hrs vary from 1.0 to 1.42. In S11c with the dose response for [L2Zn3]6+ at the lower doses the values are 1.24, 1.27, 0.83 then 0.98 and finally 0.69. Furthermore, the pS108 bands do not appear to belong to the same gel as the total bands. Overall, these experiments are not convincing.

C12 Dephosphorylation of other sites. The authors have restricted themselves to commercially available antiphosphosite antibodies which is reasonable.

C13 enhanced pT172. The authors explanation that the enhanced pT172 was due to ATP depletion is reasonable.

C14 Src response to [L2Zn3]6+ and [L2Cu3]6+ treatments. While somewhat speculative the authors have provided an acceptable explanation.

C15 Effect of [L2Zn3]6+ on glycolysis and respiration. The authors have adequately dealt with this question.

C16 The authors have provided a substantial amount of new data and addressed most of the reviewer's concerns. However, in my view the in vitro phosphatase experiments do not seem to be reliable.

Self-Assembly of an Anion Receptor with Metal-dependent Kinase Inhibition and Potent *In Vitro* Anti-Cancer Properties.

Simon J. Allison^{1*}, Jaroslaw Bryk¹, Christopher J. Clemett¹, Robert A. Faulkner¹, Michael Ginger¹, Hollie B. S. Griffiths¹, Jane Harmer¹, P. Jane Owen-Lynch¹, Emma Pinder¹, Heiko Wurdak², Roger M. Phillips^{1*} and Craig R. Rice^{1*}

Response to Reviewers comments

Reviewer #1 (Remarks to the Author):

In this work, the authors describe the synthesis and appealing anticancer activity of self-assembling trimetallic complexes, investigate their mechanism of action and identify the putative biomolecular targets, although it is not easy to figure out the mechanism and its biological targets. This work offers good enlightenment to the biological application of anion coordination chemistry. I would recommend acceptance of this manuscript after the following points are addressed.

1. What's the binding ability of the $[L_2M_3]^{6+}$ to PO_4^{3-} or SO_4^{2-} ? What's the real molecular species for $[L_2Zn_3(SO_4)]^{4+}$ or $[L_2Zn_3(O_3POPh)]^{4+}$ in the cell culture environments with high phosphate concentration (mM)?

Author response: We thank the reviewer for highlighting this important point. A competitive ³¹P NMR experiment has been carried out in response to this comment (see ESI Figure S4.6) where pre-encapsulated $[L_2Zn_3(SO_4)]^{4+}$ was reacted with NaH_2PO_4 and whilst the NMR showed some $[L_2Zn_3(PO_4)]^{3+}$ the major species was free NaH_2PO_4 indicating that, whilst some of the phosphate had displaced the sulfate, $[L_2Zn_3(SO_4)]^{4+}$ was still the major species. This is further supported by previous work (see Ref 31) which demonstrates at approximately neutral pH the capsule isn't selective to phosphate with this selectivity only induced at much higher pH. In the main paper, we have modified the text to include the following sentence: 'Studies using ³¹P NMR demonstrate that, whilst phosphate is present in the media, this anion does not entirely displace the pre-encapsulated anion and in a competitive experiment $[L_2Zn_3(SO_4)]^{4+}$ is still the major species (see ESI figure S4.6).' In the ESI, an additional figure (S4.6) has been added.

2. Some explanations should be provided. Why is it different against MCF10A cells? It is noteworthy that $[L_2Zn_3]^{6+}$ pre-encapsulated with PO_4^{3-} resulted in similar potencies, except against MCF10A cells, to $[L_2Zn_3]^{6+}$ alone. Due to significant quantities of phosphate in the cell culture media (~1mM), it is likely that $[L_2Zn_3]^{6+}$ would encapsulate PO_4^{3-} to form $[L_2Zn_3(PO_4)]^{3+}$.

Author response: Interestingly, the inorganic salt composition of the growth media for MCF10A (Eagles MEM) is different from the other cell lines (typically RPMI1640) and there are differences in the levels of phosphate and sulfate in these media. In the former case whilst the phosphate is present at lower amounts in this media, more sulfate is present, the combined effects of which may affect the speciation. However, other subtle mechanistic effects cannot be ruled out. We have modified the text to include the following sentence: 'Interestingly, however, the media used to culture MCF10A cells (Eagles MEM) contains more sulfate and lower amounts of phosphate than the culture media for most of the other cell lines and this may affect speciation and activity.'

3. Can the complexes be internalized into the cells? Some kinases locate at the cell membrane, thus the complexes may not need to enter the cells to exert the anticancer activity.

Author response: The results presented in ESI Fig. S16 suggest that the $[L_2Zn_3]^{6+}$ complex can be internalised. (see reviewer 3 C7 below). However, we cannot currently exclude the possibility that anti-cancer activity of the complexes may be exerted without the complexes entering the cells, for example by the complexes acting on kinases located at the cell membrane (e.g. receptor tyrosine kinases) affecting downstream phospho-signalling. A sentence to this effect has now been added to the manuscript: 'We cannot, however, currently exclude the possibility that anti-cancer activity of the complexes may be exerted without the complexes entering the cells, for example by the complexes transiently interacting with kinases located at the cell membrane (e.g. receptor tyrosine kinases) affecting downstream phospho-signalling.'

4. In the sentence, "Following $[L_2Zn_3]^{6+}$ treatment of HCT116 cancer cells, a small decrease (~10%) in total levels of Tyr- phosphorylated proteins was detected by immunoblotting using a pan phosphotyrosine antibody although there was also evidence of increased p-Y of some proteins (*) (Fig. 14)", how about the reliability of decrease of 10%?

Author response: Following quantification by densitometry this indicates a small decrease of around 10% in the HCT116 cells. Whilst this effect appears very small raising questions of its reliability or significance, a small decrease is consistent with the selective dephosphorylation of multiple proteins by the complex rather than global non-specific dephosphorylation. Future studies aim to understand the selectivity and to identify which proteins and phospho-sites are dephosphorylated (e.g. through an unbiased phospho-proteomics approach). The following sentence has now been added to the manuscript 'Using an unbiased phospho-proteomics approach, future studies will aim to identify which phospho-sites on proteins are selectively dephosphorylated by $[L_2Zn_3]^{6+}$.'

Reviewer #2 (Remarks to the Author):

The authors have addressed properly all the queries raised by this reviewer. Therefore, I recommend publication of this interesting manuscript.

Reviewer #3 (Remarks to the Author):

The authors have fully addressed the points raised by referee 3.

Specific points

C1 $PhOPO_3$ and phosphotyrosine dephosphorylation clarified.

C2 Clarified phosphothreonine diastereoisomers.

C3 The authors provide evidence that $[L_2Zn_3]^{6+}$ can partially dephosphorylate phosphorylated proteins thereby establishing that it works on phosphorylated amino acids within proteins. The authors claim that demonstrating that $[L_2Zn_3]^{6+}$ acted on pSer-Pro and pThr-Pro sequences was beyond the scope of the study. Given the widespread occurrence of Ser-Pro phosphorylation site targets in signaling pathways the referee considers this is still an important point and should be at least highlighted in the Discussion.

Author response: We thank the reviewer for this point and we have added the following new text to the discussion to highlight this and the need to further understand the selectivity of dephosphorylation by $[L_2Zn_3]^{6+}$, including whether it has dephosphorylation activity towards serine/threonine proline motifs. 'Using an unbiased phospho-proteomics approach, future studies will aim to identify which phospho-sites on proteins are selectively dephosphorylated by $[L_2Zn_3]^{6+}$. This will include whether $[L_2Zn_3]^{6+}$ shows any activity towards phospho-serine/threonine proline motifs, a common regulatory motif in many signalling proteins.'

C4 Experimental conditions clarified.

C5 Satisfactory explanation that HEPES does not compete.

C6 The authors need to mention "we assume the phosphorylated amino acids hydrolyse to the alcohols" in the text.

Author response: Comment added.

C7 Differential uptake. The authors have adequately investigated this point with the Zn^{2+} reporter dyeTSQ.

C8 The authors have adequately addressed the query that $[L_2Cu_3]^{6+}$ nor $[L_2Zn_3]^{6+}$ can act as ATP competitive inhibitors.

C9 The authors have adequately explained the Src differential dephosphorylation.

C10 In vitro phosphatase assay. AMPK was incubated with $[L_2Zn_3]^{6+}$ \pm AMP, ATP. In Fig S12a the control without ATP or AMP addition of $[L_2Zn_3]^{6+}$ gives an apparent increase in pT172. Across the control and + AMP the addition of $[L_2Zn_3]^{6+}$ reduces the loading control for AMPK ie possibly precipitating the AMPK. In S12b the results do not appear to show any effect of ATP and AMP on pS108 dephosphorylation. In S12c at 80 microM AMP there is less AMPK but more at 160 microM. Is AMP at 160 microM protecting the enzyme from precipitation as queried in S12a. Overall, the results do not appear to be informative.

Author response: As the reviewer suggests, it is possible that $[L_2Zn_3]^{6+}$ is causing some precipitation of AMPK resulting in the observed decrease in total AMPK levels. It is also possible that dephosphorylation by $[L_2Zn_3]^{6+}$ destabilises AMPK and increases its susceptibility to degradation. Both of these possibilities are now mentioned in the ESI text (Phosphatase assay results:) for this

supplementary figure. However, we agree that this makes interpretation of the data more difficult and effects are quite modest. Whilst fig. S12a shows some dephosphorylation of AMPK α in the presence of AMP or ATP, we agree that the data for phosphoS108 AMPK β is less convincing and have modified the text accordingly. We would be happy to remove fig. S12a and b from the supplementary material if requested as this additional data that was suggested as part of the previous round of revisions is periphery to the key findings of the manuscript.

The ESI text has been modified as follows: 'Where total levels of the AMPK subunit decreased in response to incubation with [L₂Zn₃]⁶⁺, we hypothesise that this could be due to either some precipitation of the kinase by the [L₂Zn₃]⁶⁺ complex or a consequence of dephosphorylation of the kinase by [L₂Zn₃]⁶⁺ resulting in its destabilisation and increased susceptibility to degradation.'... 'Effects of [L₂Zn₃]⁶⁺ on p-T172 and p-S108 appeared to be dependent on the presence of ATP or AMP although observed effects were quite modest (Fig. S12).'

With respect to fig. S12C (measuring AMPK kinase activity), the important and key finding from this experiment is that the presence of AMP (either at 80 or 160 microM) does not prevent inhibition of AMPK kinase activity by [L₂Zn₃]⁶⁺, with the % inhibition of AMPK activity similar in the presence of AMP (~20-30% AMPK activity remaining) as in the absence of AMP (~30% AMPK activity remaining).

The ESI text has been modified as follows: 'AMP binding to the regulatory γ AMPK subunit is known to help protect p-T172 from physiological phosphatases.^{B13} In an AMPK kinase activity assay (Promega), however, ~~Furthermore, altering increasing AMP concentration (0-160 μ M) had no adverse effect on the percentage inhibition of AMPK kinase activity by [L₂Zn₃]⁶⁺ (Fig. S12c).'~~

C11 [L₂Zn₃]⁶⁺ dose response curve and a time course. Again, in S11a there appears to be a loss of the total AMPK with [L₂Zn₃]⁶⁺ incubation. But in S11b the total beta1 incubated with [L₂Zn₃]⁶⁺ looks fine. However, the control pS108 values over the 4hrs vary from 1.0 to 1.42. In S11c with the dose response for [L₂Zn₃]⁶⁺ at the lower doses the values are 1.24, 1.27, 0.83 then 0.98 and finally 0.69. Furthermore, the pS108 bands do not appear to belong to the same gel as the total bands. Overall, these experiments are not convincing.

Although the levels of total AMPK α 1 decline with increasing incubation time with [L₂Zn₃]⁶⁺ (S11a), the decrease in phospho-Thr172 levels of AMPK α is greater than that observed for total AMPK α levels. This is consistent with dephosphorylation of phospho-Thr172 by [L₂Zn₃]⁶⁺. This is evident both from the raw blots and is confirmed by densitometric quantification and after normalisation for the variation in total AMPK α levels. We suspect that the decrease in total AMPK α 1 levels may be caused by dephosphorylation of phospho-T172 (and possible de-phosphorylation of other phospho-amino acids of AMPK α 1 that we have not probed for) resulting in destabilisation of AMPK α and its increased susceptibility to degradation. As the reviewer points out, another possibility is precipitation of AMPK α 1 by [L₂Zn₃]⁶⁺ and both of these possibilities are now mentioned in the ESI text relating to this supplementary data.

For S11b, total levels of AMPK β 1 do not decline with [L₂Zn₃]⁶⁺ incubation and it is therefore easier to see that there is a decrease in phospho-S108 AMPK β 1 with increasing time of [L₂Zn₃]⁶⁺ incubation relative to AMPK β 1 total levels. In contrast, with the DMSO control, there is no decrease in phospho-S108 AMPK β 1 with increasing time of [L₂Zn₃]⁶⁺ incubation. We would also like to point out, as is detailed in the methods section of the ESI, that the $\alpha\beta\gamma$ isoform composition of the hetero-trimeric recombinant AMPK enzyme used for S11a and S12a to analyse AMPK α 1 (and phosphoT172) and that used in S11b and S12b to analyse AMPK β 1 (and phosphoS108) are different and that these constitute

different experiments. For the data presented for AMPK α 1 (and phosphoT172), a commercially bought purified human α 1 β 2 γ 1 AMPK isoform was utilised, however, the specificity of the commercial phosphoS108 AMPK β 1 antibody against the β 1 subunit (but not the β 2 subunit) necessitated the purchase of a different human recombinant AMPK enzyme, α 2 β 1 γ 2, for analysis of phosphoS108. Further clarity about this is now provided in the S11 and S12 figure legends (highlighted in yellow) and the ESI Phosphatase assay results.

The ESI text has been modified as follows: 'For analysis of effects on pT172 AMPK α 1, purified human α 1 β 2 γ 1 AMPK kinase was used. For analysis of effects on pS108 of AMPK β 1, purified human α 2 β 1 γ 1 AMPK kinase was used. Where total levels of the AMPK subunit decreased in response to incubation with [L₂Zn₃]⁶⁺, we hypothesise that this could be due to either some precipitation of the kinase by the [L₂Zn₃]⁶⁺ complex or a consequence of dephosphorylation of the kinase by [L₂Zn₃]⁶⁺ resulting in its destabilisation and increased susceptibility to degradation.'

For S11c, the pS108 AMPK β 1 blot and the total AMPK β 1 blot are necessarily from different gels as they would be detected at the same molecular weight if analysed on the same gel, however, the samples run are from the same experiment.

C12 Dephosphorylation of other sites. The authors have restricted themselves to commercially available antiphosphosite antibodies which is reasonable.

C13 enhanced pT172. The authors explanation that the enhanced pT172 was due to ATP depletion is reasonable.

C14 Src response to [L₂Zn₃]⁶⁺ and [L₂Cu₃]⁶⁺ treatments. While somewhat speculative the authors have provided an acceptable explanation.

C15 Effect of [L₂Zn₃]⁶⁺ on glycolysis and respiration. The authors have adequately dealt with this question.

C16 The authors have provided a substantial amount of new data and addressed most of the reviewer's concerns. However, in my view the in vitro phosphatase experiments do not seem to be reliable.

Author response: Please see our detailed responses to C10 and C11. We hope that we have now addressed this.

We thank all the referees for having taken the time to thoroughly review the manuscript and the accompanying large supplementary ESI section. We hope we have now addressed all outstanding comments and provided the required clarification and that the manuscript is now acceptable for publication in Nature Communications.

With best wishes

Craig R. Rice, Simon Allison and Roger Phillips.

Reviewers' Comments:

Reviewer #1:

Remarks to the Author:

The authors have fully addressed the points raised by thre reviewer 1. I recommend the publication of this manuscript.

Reviewer #3:

Remarks to the Author:

The authors have addressed all points raised. The authors offered to remove the data in Fig S12a and b. But I strongly recommend it be included along with all the text revisions provided. While I do not think the responses to C10 and C11 are now entirely adequate the review process has run its course.